**Subject Category:**
Biology (whole organism)

bioinformatics/genetics

respiratory burst oxidase homologues, collinearity, evolution, gene duplication, NADPH oxidases

**Author for correspondence:**
Dahui Li
e-mail: dahui2@126.com

# Evolutionary and functional analysis of the plant-specific NADPH oxidase gene family in *Brassica rapa* L.

Dahui Li, Di Wu, Shizhou Li, Yu Dai and Yunpeng Cao

College of Life Science, Anhui Agricultural University, Hefei 230036, People's Republic of China

DL, 0000-0003-2050-6618

NADPH oxidases (NOXs) have been known as respiratory burst oxidase homologues (RBOHs) in plants. To characterize the evolutionary relationships and functions of RBOHs in *Brassica rapa*, 134 *RBOH* homologues were identified from 13 plant species, including 14 members (namely *BrRBOH01–14*) from *B. rapa*. There presented 47 gene-pairs among 14 *BrRBOH*s and other *RBOH*s, consisting of five pairs within *B. rapa*, and 15 pairs between *B. rapa* and *Arabidopsis thaliana*. Together with phylogenetic analysis, the results suggested that whole-genome duplication might have played an important role in *BrRBOH* gene expansion, and these duplication events occurred after the divergence of the eudicot and the monocot lineages examined. Furthermore, gene expression of *RBOH*s in both *A. thaliana* and *B. rapa* were assayed via qRT–PCR. An RBOH gene, *BrRBOH13* in *B. rapa*, was transformed into wild-type *Arabidopsis* plants. The transgenic lines with the overexpressed level of *BrRBOH13* conferred to be more tolerant to heavy metal lead (0.05 mM) than wild-type plants. Overall, this integrated analysis at genome-wide level has provided some information on the evolutionary relationships among plant-specific NOXs and the coordinated diversification of gene structure and function in *B. rapa*.

## 1. Introduction

Reactive oxygen species (ROS) have been known as one of the cellular second messengers involved in a variety of physiological processes [1–3]. Apart from aerobic metabolism pathways, ROS are mainly produced by the catalytic activity of NADPH oxidases (NOXs) [1,4,5]. Following the first identification of NADPH oxidase known as gp91$^{phox}$, which is expressed in mammalian phagocytes [1,5], considerable amounts of gp91$^{phox}$ homologues have been identified from a variety of eukaryotes, including

animals, plants and fungi, but are absent in both most unicellular eukaryotes and prokaryotes [6,7]. These gp91$^{phox}$ homologues constitute the NOX family. Overall, members of the NOX family are termed as NOX or dual oxidase (DUXO) in animals and respiratory burst oxidase homologues (RBOHs) in plants [4,6,8], respectively. With respect to their activities, animal NOXs need to be in conjunction with other cytoplasmic proteins to form a hetero-complex [5–7], whereas plant RBOHs are functional in monomer forms [4,6].

The activity modulation of NOX family members is tightly coordinated with their structures. Previous research has suggested that all NOXs share a common structure, including domains for NADPH- and FAD-binding in their C-termini and haeme-binding in their N-termini [6,7]. Additionally, RBOHs are characterized by two EF-hands in their N-termini [4]. Therefore, AtRBOHs are directly modulated by [Ca$^{2+}$] oscillations, with elevated [Ca$^{2+}$] resulting in their activation [4,6]. The activation of RBOHs is also positively regulated by a small GTPase called Rac, whose homologues in plants are termed ROPs (Rho family GTPases of plants) [9]. GTP-bound Rac can interact with the EF-hand motifs of RBOHs, leading to their activation, probably via modification of the RBOH conformation [4,6,10–12].

Consistent with the important roles of NOX-dependent ROS in cellular signal transduction [3], accumulated research has suggested that RBOHs are involved in a number of regulatory processes in plants, such as cell growth [13,14], xylem differentiation [15], responses to hormone signalling [16] and plant defence [8].

In contrast with the great progress made in dissecting the structures and activity modulation of NOXs, their evolutionary relationships remain largely unknown. In a review by Sumimoto [6], based on their similar structures and distribution only in eukaryotes, NOX, fungal ferric reductase (FRE) and ferric-chelate reductase (FRO) were proposed to compose a flavocytochrome superfamily. Furthermore, it was postulated that the NOX and FRE families had an ancestor that originated from the fusion of a bacterial di-haeme cytochrome *b* and an FNR protein. Thereafter, the ancestor of the NOXs diverged in some lineages, such as Plantae, and was lost in others (i.e. Rhizaria) during eukaryote evolution [6].

In plants, RBOH genes and their functions have been extensively characterized in *Arabidopsis* (*Arabidopsis thaliana*), rice (*Oryza sativa*) and soybean (*Glycine max*) [4,11,17,18]. Apart from those reports, the features and relationships of RBOHs in other plant species are currently uncertain. *Brassica rapa* L. is an economically important crop. As the RBOHs have been known to be active not only across the plant growth and development, but also involved in plant defence [3,8], studying the function of RBOH genes in *B. rapa* should be beneficial to improving its cultivation and yield. In the present research, a comprehensive analysis of evolution and function of RBOHs was carried out at the whole-genome level. All RBOH protein-coding nucleotide sequences were retrieved from *B. rapa* and 12 other plant species with available genome data. The phylogenetic, gene structures and protein motifs of the identified *RBOH* genes, together with collinearity analysis on these genes, provided some clues to the evolutionary relationships among these plant-specific NOXs. Furthermore, the transgenic *Arabidopsis* plants overexpressing an RBOH gene, *BrRBOH13* in *B. rapa*, showed an enhanced tolerance to heavy metal lead, compared with the wild-type plants, inferring that the gene *BrRBOH13* may be an ideal candidate for crop improvement in *B. rapa*.

# 2. Results

## 2.1. Identification and analysis of *RBOH* homologues in plants

A total of 138 candidate *RBOH* sequences were initially identified from the 13 species examined, using the RBOH HMMs from the Pfam database and searching against the genome data. Four of the 138 identified NOXs were removed from the initial collection because they are different transcripts from one gene. The remaining NOX genes (134) were retained as *RBOH* homologues and named after their individual chromosomal locations (electronic supplementary material, table S1). The number of *RBOH*s was varied among these species. Five and eight *RBOH*s were identified in the moss *Physcomitrella patens* (*PpRBOH01–05*) and the lycophyte *Selaginella moellendorffii* (*SmRBOH01–08*), respectively. The number of the NOX family members was varied in the angiosperm species examined, including the dicotyledonous angiosperms *G. max* (17, *GmRBOH01–17*), *B. rapa* (14, *BrRBOH01–14*), *A. thaliana* (10, *AtRBOH01–10*), *Populus trichocarpa* (10, *PtRBOH01–10*), *Citrus sinensis* (8, *CsRBOH01–08*) and *Cucumis sativus* (7, *CuRBOH01–07*), and the monocotyledonous angiosperms *Zea mays* (15, *ZmRBOH01–15*), *Setaria italica* (13, *SiRBOH01–13*), *Brachypodium distachyon* (9, *BdRBOH01–09*), *O. sativa* (9, *OsRBOH01–09*) and *Sorghum bicolor* (9, *SbRBOH01–09*). These RBOH

**Figure 1.** Genomic distribution of *RBOH* genes on *B. rapa* chromosomes. Chromosomal locations of *RBOH*s are illustrated based on the physical position of each gene. The number of chromosomes is indicated on the top of each chromosome. The centromeres are marked by red ovals, respectively.

genes were unevenly distributed on individual genomes (electronic supplementary material, table S1). In *B. rapa*, 14 *BrRBOH*s were localized on chromosomes 1 (*BrRBOH01*), 2 (*BrRBOH02–04*), 3 (*BrRBOH05* and *06*), 5 (*BrRBOH07*), 6 (*BrRBOH08–10*), 9 (*BrRBOH11–13*) and 10 (*BrRBOH14*) (figure 1). Among the examined species, *G. max* had the most *RBOH* members (17) and the most even distribution across the chromosomes, with one gene each on chromosomes 1, 3, 4, 6, 7, 9–11, 15, 17, 19 and 20, two genes on chromosome 8 and three genes on chromosome 5 (electronic supplementary material, table S1).

## 2.2. Phylogenetic and gene structural analysis of the *RBOH*s

To investigate the phylogenetic relationships and molecular evolutionary history of the sequences in the 13 examined species, following the alignment of 134 RBOH proteins (electronic supplementary material, figure S1), a phylogenetic analysis was conducted and a phylogenetic tree was generated using the NJ method (electronic supplementary material, figure S2). The NJ tree showed that the 134 sequences were clustered into seven main subgroups, among which the highest numbers of members were 37 and 32 in subgroups III and I, followed by 23, 22, 10, 6 and 4 in subgroups IV, II, V, VI and VII, respectively. As shown in electronic supplementary material, figure S2 and table S2, the RBOHs from individual angiosperm species were grouped into different clades rather than a single one. Additionally, their numbers varied within different subgroups. Out of 14 BrRBOHs, half of them (BrRBOH01, 04–07, 10 and 14) were located in subgroup II, two in subgroups III (BrRBOH02 and 09), IV (BrRBOH03 and 11) and VII (BrRBOH08 and 12) and one (BrRBOH13) in subgroup I, respectively. Ten AtRBOHs were dispersed across subgroups I–IV and VI with one to four members per subgroup. The other angiosperms species also exhibited similar patterns in their RBOH distributions (electronic supplementary material, figure S2). Additionally, subgroup III contained all of the members from both the moss *P. patens* and the lycophyte *S. moellendorffii*. Moreover, the five moss members (PpRBOH01–05) and eight lycophyte members (SmRBOH01–08) were grouped into two distinct branches separated from RBOH homologues from the other species examined (electronic supplementary material, figure S2). Based on sequence alignment, it was found that all of four characteristic motifs of RBOH family (NADPH_Ox, Ferric_reduct, FAD_binding_8 and NAD_binding_6) were presented within 124 out of 134 RBOHs, including eight BrRBOH members (electronic supplementary material, figures S1 and S2). Among the remaining 10 RBOHs, BrRBOH04, BrRBOH07 and BrRBOH10 have three (NADPH_Ox, Ferric_reduct and NAD_binding_6 in RBOH04 and BrRBOH10, or NADPH_Ox, FAD_binding_8, and NAD_binding_6 in BrRBOH07), BrRBOH08, PpRBOH02 and ZmRBOH12 have two (both NADPH_Ox and NAD_binding_6 in BrRBOH08, or NADPH_Ox and Ferric_reduct in PpRBOH02 and ZmRBOH12), BrRBOH01 and BrRBOH12, CsRBOH08, and ZmRBOH08

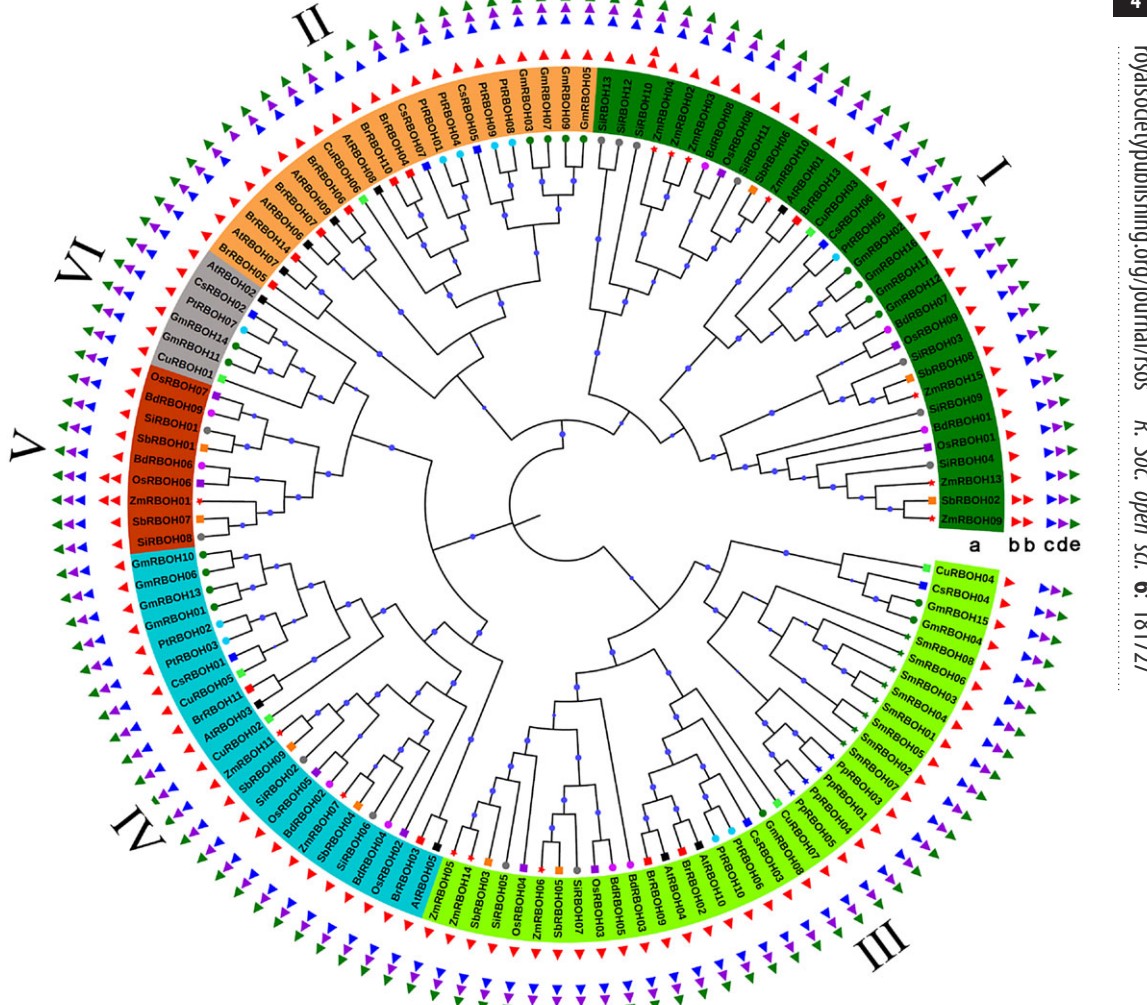

**Figure 2.** Phylogenetic relationships of RBOHs. The phylogenetic tree (circle a) is constructed, after seven members with one or two RBOH-characteristic motifs were excluded from the 134 identified RBOHs. Six subgroups are indicated with I–VI, respectively. Triangles in red, blue, purple and green colour indicate domains of NADPH_Ox, Ferric_reduct, FAD_binding_8 and NAD_binding_6, respectively. Circles b, c, d and e are composed of four above-mentioned domains within each RBOHs, respectively. Circles at the individual nodes represent bootstrap support.

have one (NADPH_Ox) of RBOH-characteristic motifs, respectively (electronic supplementary material, figure S2). Together with BrRBOH01, PpRBOH02 and ZmRBOH12, four members of the subgroup VII (BrRBOH08, BrRBOH12, CsRBOH08 and ZmRBOH08) were excluded from the 134 identified RBOHs and the construction of an unrooted phylogenetic tree using RAxML (figure 2), because of their great divergence with other RBOH members. As shown in figure 2, similar division of subgroups (I–VI) was built, compared with the previous one (electronic supplementary material, figure S2). And there showed a high conservation of these characteristic motifs among the RBOHs identified (figure 3).

To characterize their gene structural diversity, the exon–intron organizations of the *RBOH*s were analysed (figure 4*a*). The number of exons was diverse, with a minimum of three (i.e. *BrNADPH12* and *PpNADPH02*) and a maximum of 15 (*CuRBOH05*, *OsRBOH04*, *SbRBOH03*, *SiRBOH09*, *SmRBOH05* and *SmRBOH08*). Coordinated with their distribution across various subgroups in the phylogenetic tree, exons of 14 *BrRBOH*s were ranged from 3 to 14. However, the gene structures within each subgroup showed a similar pattern, which supported their phylogenetic relationships. In addition, using the program MEME, it was demonstrated that most representatives of the motifs in RBOH proteins from the same subgroup showed a conservation in both motif distribution and composition (figure 4*b*). Although amino acid sequences were highly conserved throughout RBOH members of individual subgroups within each of these motifs (electronic supplementary material, figure S1), the similarity at the whole sequence level was 19.33% among these RBOHs. Additionally,

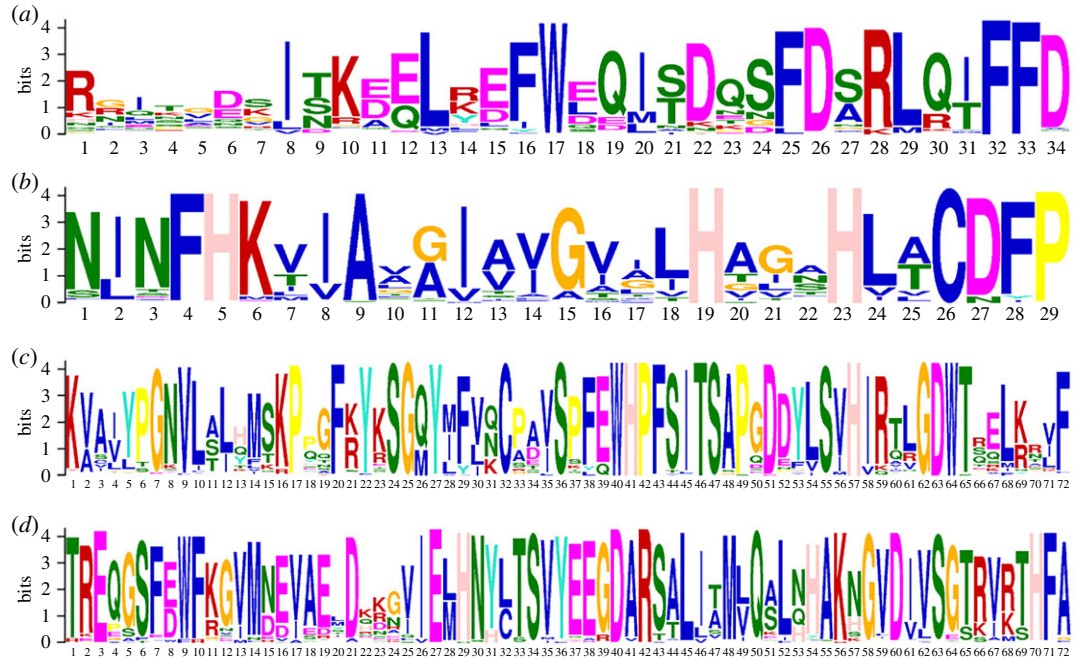

**Figure 3.** Sequence logos of the conserved motifs of NADPH_Ox (*a*), Ferric_reduct (*b*), FAD_binding_8 (*c*) and NAD_binding_6 (*d*) among RBOH proteins. The bit score indicates the information content for each position in the sequence. The height of each character is correlated to the amino acid conservation across all the proteins tested.

the results from PIECE analysis supported the results of MEME analysis (electronic supplementary material, figure S3).

## 2.3. Collinearity analysis of *RBOH*s

It is thought that gene duplication is the main factor that promotes the evolution of multi-gene families. Collinearity analysis on gene sequences flanking the chromosomal regions containing *RBOH* genes was carried out to unravel their duplication types. As a result, a total of 47 gene-pairs with segmental duplications were identified among 14 *BrRBOH*s and other *RBOH*s, consisting of five intraspecies-pairs within *B. rapa* and 42 interspecies-pairs across *B. rapa* and other species (table 1). Among 42 gene-pairs with interspecies collinearity (table 1), there existed 15, 17, 5 and 5 pairs between *B. rapa* and *A. thaliana*, *G. max*, *C. sinensis* and *P. trichocarpa*, respectively. It is noticeable that there is no gene-pair with segmental duplication among *B. rapa* and moss *P. patens*, lycophyte *S. moellendorffii* or the monocotyledonous species examined. Pictorial microsynteny demonstrated five gene-pairs (*BrRBOH01/BrRBOH06*, *BrRBOH02/ BrRBOH09*, *BrRBOH04/BrRBOH10*, *BrRBOH05/BrRBOH14*, *BrRBOH13/BrRBOH14*) within *B. rapa*, and 15 gene-pairs (*BrRBOH01/AtRBOH06*, *BrRBOH01/AtRBOH09*, *BrRBOH02/AtRBOH04*, *BrRBOH02/ AtRBOH10*, *BrRBOH03/AtRBOH05*, *BrRBOH04/AtRBOH08*, *BrRBOH05/AtRBOH07*, *BrRBOH06/ AtRBOH06*, *BrRBOH06/AtRBOH09*, *BrRBOH08/AtRBOH02*, *BrRBOH09/AtRBOH10*, *BrRBOH10/ AtRBOH08*, *BrRBOH11/AtRBOH03*, *BrRBOH13/AtRBOH01*, *BrRBOH14/AtRBOH07*) between *B. rapa* and *A. thaliana* (figures 5 and 6), due to segmental duplications. Apart from *BrRBOH*s, gene collinearity with segmental duplication types were also identified in *RBOH*s among other species examined (electronic supplementary material, table S2). To assess the evolutionary rates among these gene-pairs, the *Ka* (non-synonymous nucleotide substitutions) to *Ks* (synonymous nucleotide substitutions) ratios were calculated (table 1; electronic supplementary material, table S2). The *Ka/Ks* values ranged from 0.042 to 0.881 for the gene-pairs with segmental duplications between *B. rapa* and other species, while from 0.120 to 0.566 for those within *B. rapa* (table 1).

To further investigate the functional divergence of amino acid sequences after RBOH duplication, a representative subgroup (III) in the phylogenetic tree was selected because it consisted of various plant lineages. Subgroup III was classified into three clusters: an ancient one with five, eight and four members from unicellular moss, lycophyte and eudicot species, respectively; an intermediate cluster with nine eudicot members, including two *B. rapa* RBOHs (BrRBOH02 and 09); and a modern cluster with 11 monocot members (electronic supplementary material, figure S4). Through analysis of type I

(a)   (b)

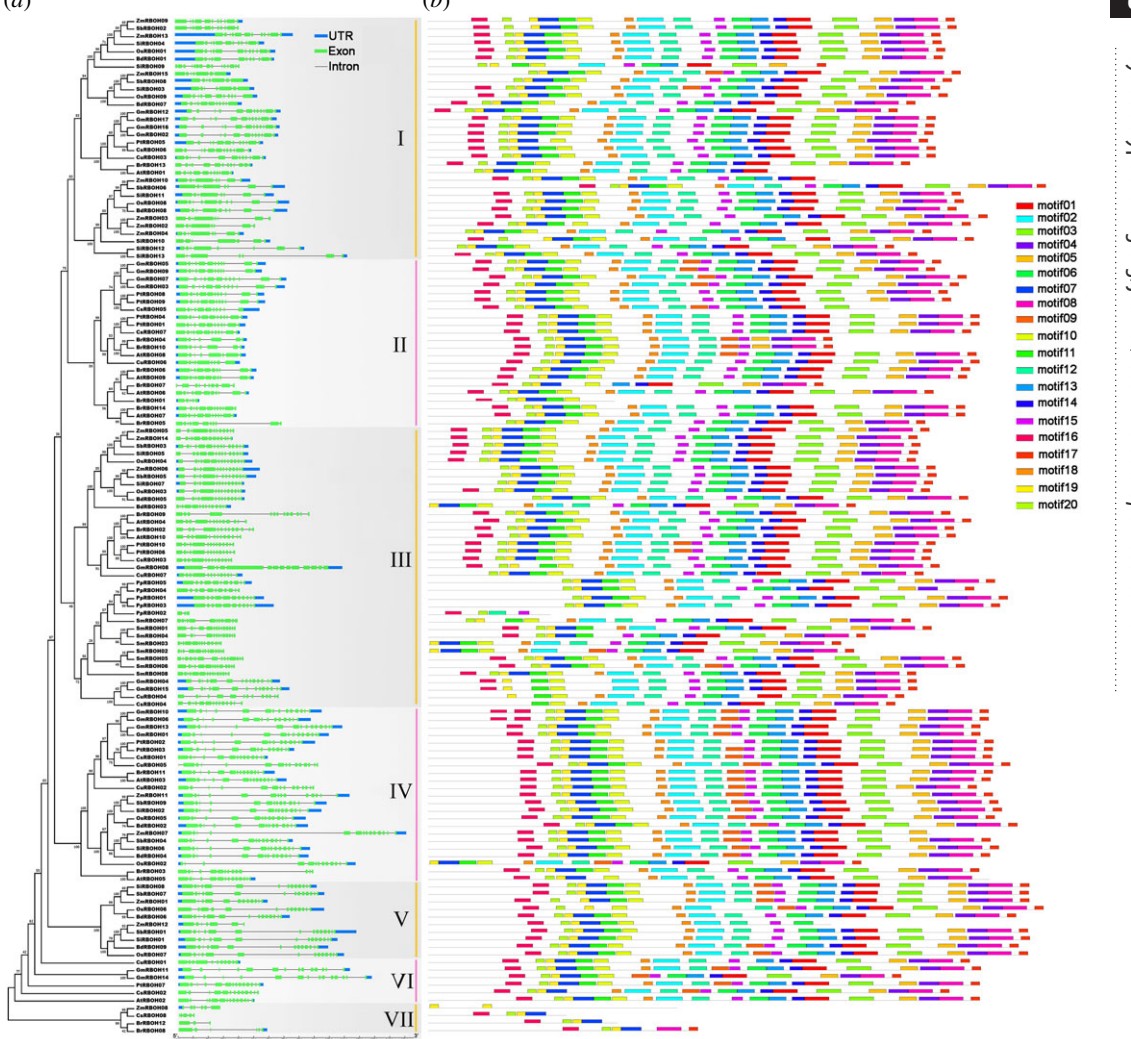

**Figure 4.** Gene structures of *RBOH*s (*a*). The vertical lines indicate the seven corresponding gene subgroups. Blue boxes, 5′ or 3′ untranslated region (UTR); green boxes, exons; black lines, introns. Box and line lengths are scaled based on gene length. Protein motifs of RBOHs (*b*). Motifs are illustrated using MEME program. The vertical lines indicate the seven corresponding gene subgroups. Boxes in different colours indicate 20 different motifs.

functional divergence, five and three pivotal amino acid positions were identified within the ancient–intermediate and ancient–modern cluster pairs, respectively (table 2). Conversely, no pivotal amino acid positions were found from the corresponding cluster pairs with type II functional divergence, or from the intermediate–modern pair with type I divergence (table 2).

## 2.4. Quantitative analysis of *RBOH* expression in *Arabidopsis* and *B. rapa*

The expression patterns of the *RBOH*s in different tissues (roots, hypocotyls, stems, leaves and flowers) were examined using qRT–PCR. In *Arabidopsis*, irrespective of their expression levels, all of the *AtRBOH*s were constitutively expressed with various patterns in the different tissues examined (figure 6). The expression levels of *AtRBOH01*, *06* and *09* were higher in roots than in other tissues, while *AtRBOH02* was expressed at the highest levels in both roots and leaves, followed by those in hypocotyls. Four other *AtRBOH* genes, *AtRBOH03*, *05*, *07* and *08*, showed the closing expression levels in a broader range of tissues (figure 6), with the relatively higher expression in roots (*AtRBOH03*, *05* and *07*) and stems (*AtRBOH08*). In addition to roots, *AtRBOH03* was highly expressed in hypocotyls and stems, as well *AtRBOH05* and *07* in flowers. The remaining genes, *AtRBOH04* and *10*, showed a unique expression pattern, with expression peaks in flowers, compared with the expression levels in other tissues (figure 6).

**Table 1.** Collinearity relationship of *RBOH* genes among *B. rapa* and other species examined.

| synteny sequence 1 | chromosome | synteny sequence 2 | chromosome | duplication type | *Ka/Ks* | purifying selection |
|---|---|---|---|---|---|---|
| *BrRBOH01* | Chr1 | *BrRBOH06* | Chr3 | segmental | 0.252 | yes |
| *BrRBOH02* | Chr2 | *BrRBOH09* | Chr6 | segmental | 0.129 | yes |
| *BrRBOH04* | Chr2 | *BrRBOH10* | Chr6 | WGD | 0.566 | yes |
| *BrRBOH05* | Chr3 | *BrRBOH14* | Chr10 | WGD | 0.468 | yes |
| *BrRBOH13* | Chr9 | *BrRBOH14* | Chr10 | segmental | 0.120 | yes |
| *BrRBOH01* | Chr1 | *AtRBOH06* | Chr4 | WGD | 0.371 | yes |
| *BrRBOH01* | Chr1 | *AtRBOH09* | Chr5 | WGD | 0.299 | yes |
| *BrRBOH02* | Chr2 | *AtRBOH04* | Chr3 | WGD | 0.117 | yes |
| *BrRBOH02* | Chr2 | *AtRBOH10* | Chr5 | WGD | 0.099 | yes |
| *BrRBOH03* | Chr2 | *AtRBOH05* | Chr4 | WGD | 0.215 | yes |
| *BrRBOH04* | Chr2 | *AtRBOH08* | Chr5 | WGD | 0.093 | yes |
| *BrRBOH05* | Chr3 | *AtRBOH07* | Chr5 | WGD | 0.439 | yes |
| *BrRBOH06* | Chr3 | *AtRBOH06* | Chr4 | WGD | 0.137 | yes |
| *BrRBOH06* | Chr3 | *AtRBOH09* | Chr5 | WGD | 0.104 | yes |
| *BrRBOH08* | Chr6 | *AtRBOH02* | Chr1 | WGD | 0.881 | yes |
| *BrRBOH09* | Chr6 | *AtRBOH10* | Chr5 | WGD | 0.122 | yes |
| *BrRBOH10* | Chr6 | *AtRBOH08* | Chr5 | WGD | 0.096 | yes |
| *BrRBOH11* | Chr9 | *AtRBOH03* | Chr1 | WGD | 0.067 | yes |
| *BrRBOH13* | Chr9 | *AtRBOH01* | Chr1 | WGD | 0.170 | yes |
| *BrRBOH14* | Chr10 | *AtRBOH07* | Chr5 | WGD | 0.201 | yes |
| *BrRBOH01* | Chr1 | *GmRBOH03* | Chr4 | segmental | 0.538 | yes |
| *BrRBOH01* | Chr1 | *GmRBOH07* | Chr6 | segmental | 0.524 | yes |
| *BrRBOH01* | Chr1 | *GmRBOH09* | Chr8 | segmental | 0.596 | yes |
| *BrRBOH06* | Chr3 | *GmRBOH03* | Chr4 | segmental | 0.283 | yes |
| *BrRBOH06* | Chr3 | *GmRBOH07* | Chr6 | segmental | 0.214 | yes |
| *BrRBOH06* | Chr3 | *GmRBOH09* | Chr8 | segmental | 0.316 | yes |
| *BrRBOH06* | Chr3 | *GmRBOH14* | Chr17 | segmental | 0.532 | yes |
| *BrRBOH11* | Chr9 | *GmRBOH01* | Chr1 | segmental | 0.192 | yes |
| *BrRBOH11* | Chr9 | *GmRBOH06* | Chr5 | segmental | 0.197 | yes |
| *BrRBOH11* | Chr9 | *GmRBOH10* | Chr8 | segmental | 0.131 | yes |
| *BrRBOH11* | Chr9 | *GmRBOH13* | Chr11 | segmental | 0.122 | yes |
| *BrRBOH13* | Chr9 | *GmRBOH02* | Chr3 | segmental | 0.252 | yes |
| *BrRBOH13* | Chr9 | *GmRBOH12* | Chr10 | segmental | 0.258 | yes |
| *BrRBOH13* | Chr9 | *GmRBOH16* | Chr19 | segmental | 0.256 | yes |
| *BrRBOH14* | Chr10 | *GmRBOH03* | Chr4 | segmental | 0.342 | yes |
| *BrRBOH14* | Chr10 | *GmRBOH05* | Chr5 | segmental | 0.358 | yes |
| *BrRBOH14* | Chr10 | *GmRBOH09* | Chr8 | segmental | 0.318 | yes |
| *BrRBOH02* | Chr2 | *CsRBOH03* | scaffold00014 | segmental | 0.176 | yes |
| *BrRBOH06* | Chr3 | *CsRBOH05* | scaffold00087 | segmental | 0.124 | yes |
| *BrRBOH09* | Chr6 | *CsRBOH03* | scaffold00014 | segmental | 0.293 | yes |
| *BrRBOH11* | Chr9 | *CsRBOH01* | scaffold00009 | segmental | 0.042 | yes |

(*Continued.*)

| synteny sequence 1 | chromosome | synteny sequence 2 | chromosome | duplication type | Ka/Ks | purifying selection |
|---|---|---|---|---|---|---|
| BrRBOH14 | Chr10 | CsRBOH05 | scaffold00087 | segmental | 0.140 | yes |
| BrRBOH06 | Chr3 | PtRBOH08 | Chr12 | segmental | 0.118 | yes |
| BrRBOH10 | Chr6 | PtRBOH01 | Chr1 | segmental | 0.199 | yes |
| BrRBOH10 | Chr6 | PtRBOH04 | Chr3 | segmental | 0.217 | yes |
| BrRBOH11 | Chr9 | PtRBOH02 | Chr1 | segmental | 0.073 | yes |
| BrRBOH11 | Chr9 | PtRBOH03 | Ch3 | segmental | 0.067 | yes |

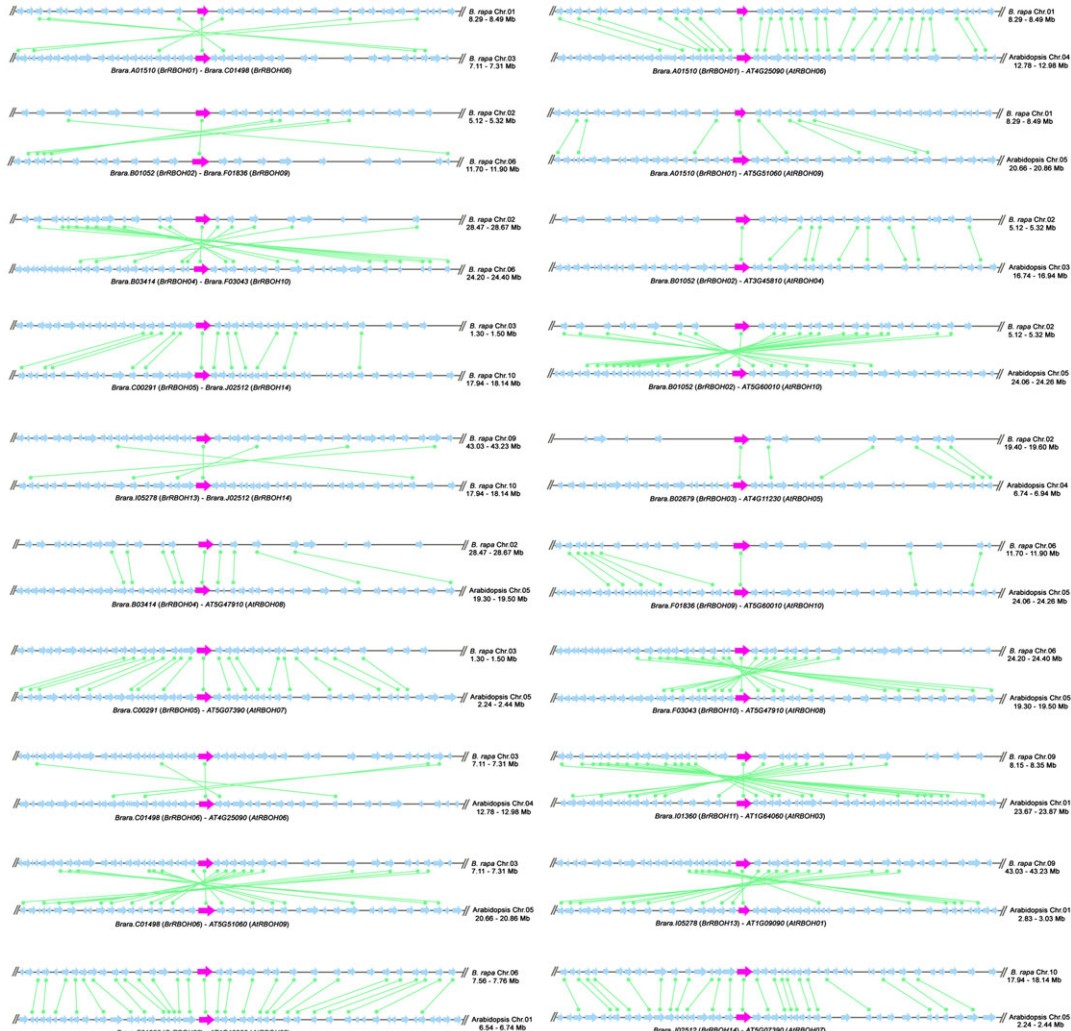

**Figure 5.** Segmental duplications among *RBOH*s in *B. rapa*, or between *B. rapa* and *Arabidopsis* by collinearity analysis, respectively. A region of 100 kb flanking at both sides of *RBOH*s (pink arrowheads) is illustrated in the figure. The black horizontal lines represent the chromosome segments, while blue arrowheads represent the individual genes. Homologous gene-pairs are connected with green lines attached by apical dots. *RBOH* gene-pairs with segmental duplications are listed under the chromosome segments.

In *B. rapa*, most of *BrRBOH*s (10 out of 14, including *BrRBOH01–04, 06–09, 11* and *12*) were expressed with the similar patterns in both vegetative organs and flowers, among which no transcriptional expression of *BrRBOH04* was found in the examined tissues (figure 6). Except for no expression in stems, *BrRBOH05* had higher levels in leaves than those in roots, hypocotyls and flowers. With higher levels in hypocotyls, leaves and flowers, and lower ones in stems, the expression pattern of *BrRBOH10* showed a

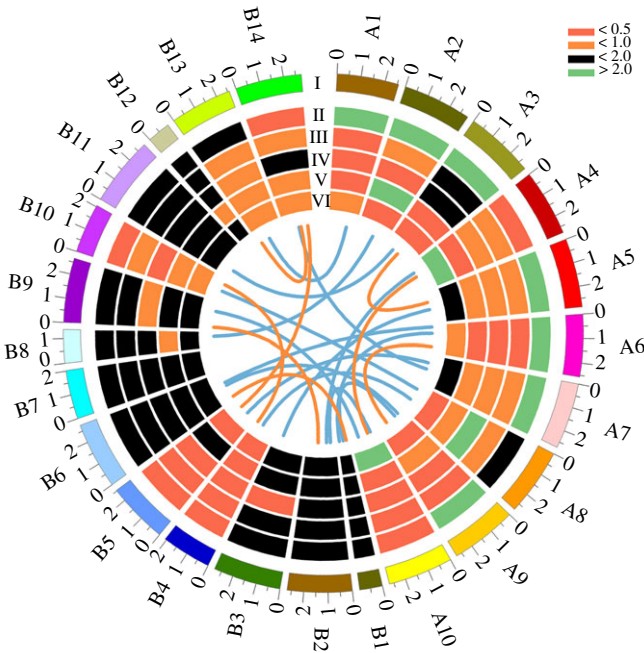

**Figure 6.** Collinearity relationships of *RBOH* members and their respective expression patterns in *Arabidopsis* and *B. rapa*. I, Ten *AtRBOH*s and 14 *BrRBOH*s are marked with A1–A10 and B1–B14, respectively. Orange linkages refer to intraspecies gene duplications. Blue linkages indicate gene duplications between *Arabidopsis* and *B. rapa*. II–VI, qRT–PCR analysis of expression profiles of *AtRBOH*s and *BrRBOH*s in roots, hypocotyls, stems, leaves and flowers, respectively. Colour bars: relative abundance of gene transcript using tubulin as the internal control. The data shown are means ($\pm$ standard deviation) of three experiments.

**Table 2.** Analysis of functional divergence. s.e., standard error; LRT, value of likelihood ratio test.

| types | cluster 1 | cluster 2 | $\Theta \pm$ s.e. | LRT | $Q_k > 0.9$ |
|---|---|---|---|---|---|
| I | ancient | intermediate | 0.488 $\pm$ 0.057 | 15.895 | 614, 631,687,694,722 |
| | ancient | modern | 0.421 $\pm$ 0.168 | 1.883 | 614, 615, 722 |
| | intermediate | modern | 0.195 $\pm$ 0.144 | 1.825 | n.a. |
| II | ancient | intermediate | 0.987 $\pm$ 0.238 | n.a. | n.a. |
| | ancient | modern | 1.073 $\pm$ 0.259 | n.a. | n.a. |
| | intermediate | modern | 0.873 $\pm$ 0.130 | n.a. | n.a. |

reverse trend to that of *BrRBOH14* (figure 6). A unique expression pattern was found in *BrRBOH13* which showed the root-specific higher expression levels compared with other tissues (figure 6).

## 2.5. Transgenic *Arabidopsis* plants overexpressing *BrRBOH13*

Based on the phylogenetic and gene expression analysis, *BrRBOH13* was selected and transformed into the wild-type *Arabidopsis* to investigate its function in association with environmental resilience. *GUS* reporter gene in the stable *BrRBOH13* transgenic plants was constitutively expressed throughout the plant tissues (figure 7*a,b*). Furthermore, it was found that *BrRBOH13* was ectopically overexpressed in the transgenic *Arabidopsis* with its expression levels 3.1-fold higher on average than those in the wild-type (figure 7*c*).

These *BrRBOH13* transgenic lines were treated with both abiotic stresses of heavy metal lead and salt NaCl, respectively. As shown in figure 7*d*–*g*, the transgenic *Arabidopsis* plants overexpressing *BrRBOH13* have increased tolerance to heavy metal lead. Upon 0.05 mM $Pb^{2+}$ treatments, the *BrRBOH13* transgenic plants were able to keep normal growth throughout the duration from 2 to 7 days (figure 7*d,e*), whereas wild-type *Arabidopsis* has gradually undergone chlorosis and turned into senescent phenotype (figure 7*f,g*). After the treatment with salt stress (200 mM NaCl) for 7 days, the transgenic plants

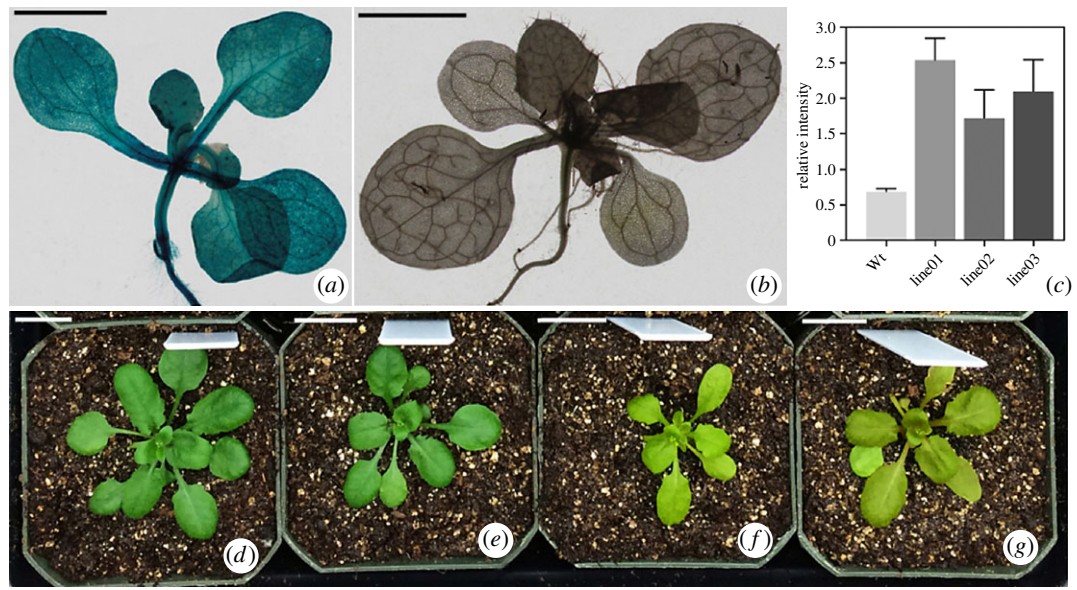

**Figure 7.** The transgenic *Arabidopsis* plants overexpressing *BrRBOH13*. GUS reporter gene expression in *BrRBOH13* transgenic line (*a*). Wild-type plant is used as negative control (*b*). qRT−PCR analysis of expression pattern of *BrRBOH13* in wild-type (Wt) and transgenic plants (line01−03) (*c*). Densitometric analysis gives quantitative measurements of gene expression, using tubulin as the internal control. The data shown are means ($\pm$ standard deviation) of three experiments. Phenotypes of the transgenic *Arabidopsis* overexpressing *BrRBOH13* (*d,e*) or the wild-type lines (*f,g*), under the $Pb^{2+}$-treatments (0.05 mM) for 2 days (*e,f*) or 7 days (*d,g*). Bars, 2 mm in (*a,b*); 1 cm in (*d−g*), respectively.

showed the appearance of both chlorosis and necrotic lesions on their leaves, compared to the control ones without NaCl-treatment (electronic supplementary material, figure S5). This result suggested that no enhanced salt tolerance was activated in *BrRBOH13* transgenic *Arabidopsis* plants.

## 3. Discussion

NOXs catalyse ROS formation and play multiple roles in development, signal transduction, defence and hormone responses [1,3,8,13,16]. While absent in prokaryotes and most unicellular eukaryotes, NOXs are widely present in fungi, plants and animals [6,7]. These NOXs constitute the NOX multi-gene family. In plants, the NOX family members are designated RBOHs [4,6]. In the present research, 134 candidate *NOXs* were identified from the species examined. Among these identified *RBOH*s, the maximal number of members was found in *G. max* (17), followed by *Z. mays* (15) and *B. rapa* (14), while the minimal number in the moss *P. patens* (5). And, the numbers of RBOHs in *Arabidopsis*, *P. trichocarpa*, rice and soya bean were consistent with those in the previous reports [4,17,18]. Inconsistency in gene numbers was found within four species, i.e. *Z. mays* (15), *S. bicolor* (9), *P. patens* (5) and *S. moellendorffii* (8) in the presented study, which were 12, 7, 4 and 5 in the previous work, respectively [17]. After rescreening of the respective genome data of these four species, all of the RBOHs in the present study have been validated with the presence of the conserved RBOH domains (electronic supplementary material, figure S1; figure 2).

As per the phylogenetic tree (figure 2; electronic supplementary material, figure S2), 14 BrRBOHs were dispersed into the subgroups II (BrRBOH01, 04−07, 10 and 14), III (BrRBOH02 and 09), IV (BrRBOH03 and 11) and VII (BrRBOH08 and 12) and I (BrRBOH13), respectively. And, each subgroup contains RBOHs from both dicotyledonous and monocotyledonous species, indicating that these *RBOH*s have undergone different expansion events because of the divergence of plant lineages examined. The relationships of 134 RBOHs indicated by the phylogenetic tree were further supported by the similar gene structure and protein-motif patterns within each subgroup. It is noticeable that although both of five moss members (PpRBOH01−05) and eight lycophyte ones (SmRBOH01−08) were grouped into subgroup III with some RBOHs, such as BrRBOH02 and BrRBOH09, they were exclusively clustered together into two distinct subclades containing no other RBOHs. This result suggests that the RBOHs from the moss and the lycophyte have closer relationships with each other than with those from other lineages. In addition, compared with the previous analysis on RBOHs among soya bean or rice and other plants [17,18], some similar phylogenetic relationships were obtained. For example, four GmRBOHs (02, 16, 17 and 12) were not only formed into

two paralogous pairs (GmRBOH02–GmRBOH16 and GmRBOH17–GmRBOH12), but also grouped with AtRBOH01 and three OsRBOHs (01, 09 and 08) into the division of subgroup I, while another four GmRBOHs (03, 07, 09 and 05), producing two paralogous pairs (GmRBOH03–GmRBOH07 and GmRBOH09–GmRBOH05), were grouped with four AtRBOHs (06, 09, 07, 08) into the division of subgroup II in the present study (figure 2).

It has been reported that large-scale gene duplication probably occurred at least four times during the divergence of eudicots and monocots about 100–200 Ma [19–21]. To uncover the contribution of gene duplications to the evolution of the RBOHs in the examined species, we assessed the gene expansion patterns. Based on the criterion (i.e. if there were three gene-pairs at least within 100 kb sequences located both down- and upstream of a pair of RBOH genes, the two chromosomal blocks were considered as segmental duplications), collinearity analysis showed that there presented 47 gene-pairs among 14 BrRBOHs and other RBOHs, including 5 and 42 pairs with intraspecies and interspecies collinearity, respectively (table 1). It has been reported that B. rapa genome had undergone a complex history of evolution in Brassicaceae, including a unique whole-genome triplication (WGT) event [22]. The gene-pairs demonstrated in figure 5 and table 1, including two gene-pairs within BrRBOHs (BrRBOH04/BrRBOH10, BrRBOH05/BrRBOH14) and 15 gene-pairs between BrRBOHs and AtRBOHs, are representatives on the synteny blocks of WGT according to the evolution of B. rapa genome [22]. Three of five pairs within BrRBOHs, i.e. BrRBOH01/BrRBOH06, BrRBOH02/BrRBOH09 and BrRBOH13/BrRBOH14, were from segmental duplications (table 1). These results suggest that whole-genome duplication (WGD) might have played an important role in BrRBOH gene expansion, leading to structural and functional novelty that increased tolerance to abiotic and biotic pressures during evolution. Furthermore, according to the analysis of tandem genes and their chromosomal locations, the majority of RBOHs were randomly distributed across individual genomes (electronic supplementary material, table S1), indicating the segmental duplication of these RBOHs that occurred was in accordance with the large-scale duplication events. As the collinearity relationships between BrRBOHs and other RBOHs have merely been found from the dicotyledonous species examined, it was deduced that segmental duplication events occurred after the divergence of the eudicots and the monocots.

Moreover, within the phylogenetic tree (figure 2), the majority of RBOHs located on the same chromosome from a specific species did not cluster into the same subgroup. For instance, AtRBOH01, AtRBOH02 and AtRBOH03 on Arabidopsis chromosome 1 were included in subgroups I, VI and IV, and BrRBOH02, 03 and 04 on B. rapa chromosome 2 were in subgroups III, IV and II, respectively, indicating the further diversification of RBOHs among the plant lineages examined during evolution.

The Ka/Ks ratio is considered an indicator for determining the type of selection pressure [23]. In the present research, all of the evaluated Ka/Ks ratios between gene-pairs with collinearity relationship were less than 1 (table 1; electronic supplementary material, table S2), although the ratios were varied, indicating these genes have undergone purifying selection to different extents since duplication. Furthermore, according to previous reports [20,21], type I functional divergence is tightly associated with the evolutionary rate of a duplicated gene, possibly leading to a change in gene function, whereas type II divergence is associated with the physico-chemical features of a sequence. In the light of the presence of pivotal amino acid substitutions between ancient and intermediate or modern cluster pairs with type I functional divergence (table 2), it is likely that diversification occurred after RBOH gene duplication.

To investigate the functional diversity of the RBOHs, both B. rapa and the model species Arabidopsis were assayed for their RBOH expression patterns. Based on the respective expression data, AtRBOHs are constitutively expressed throughout the plant development. Previous research has suggested that AtRBOHs can be classified into three basic types: (i) AtRBOH D (i.e. AtRBOH08) and F (AtRBOH03), which are widely expressed throughout the plant; (ii) AtRBOH A (AtRBOH07), B (AtRBOH01), C (AtRBOH09), E (AtRBOH02), G (AtRBOH06) and I (AtRBOH05), which are expressed in the roots; and (iii) AtRBOHs H (AtRBOH10) and J (AtRBOH04), which are expressed in pollen [4]. In the present research, the expression patterns of the 10 AtRBOHs according to data from the qRT–PCR were consistent with the previous report.

In B. rapa, 14 BrRBOHs were classified into two types according to expression clustering: (i) 10 members were expressed with similar levels in the vegetative organs and flowers examined; (ii) the remaining four exhibited the various trends in expression, one of which was the higher expression levels of BrRBOH13 in roots than in other organs. In addition to being grouped into the same cluster within the BrRBOH phylogenetic tree, BrRBOH13 and AtRBOH01 had a collinearity relationship with each other. Moreover, BrRBOH13 showed a less homologous relationship with other BrRBOH members, indicating it might be involved in distinct functions. Previously, some AtRBOHs have been found to be associated with abiotic or biotic factors [8,13,16]. In the present research, the BrRBOH13 transgenic Arabidopsis plants were

**Table 3.** Plant species selected for identification of *RBOH* genes.

| plant lineage | species | website for genome data |
| --- | --- | --- |
| unicellular moss | *Physcomitrella patens* | http://genome.jgi-psf.org/ |
| lycophyte | *Selaginella moellendorffii* | http://genome.jgi-psf.org/ |
| eudicots | *Arabidopsis thaliana* | http://www.arabidopsis.org/ |
| | *Brassica rapa* | http://brassicadb.org/brad/ |
| | *Citrus sinensis* | http://www.phytozome.org/ |
| | *Cucumis sativus* | http://genome.jgi-psf.org/ |
| | *Glycine max* | http://genome.jgi-psf.org/soybean/ |
| | *Populus trichocarpa* | http://genome.jgi-psf.org/ |
| monocots | *Brachypodium distachyon* | http://www.brachypodium.org |
| | *Oryza sativa* | http://rapdb.dna.affrc.go.jp/ |
| | *Setaria italica* | http://www.phytozome.org |
| | *Sorghum bicolor* | http://genome.jgi-psf.org/ |
| | *Zea mays* | http://www.maize-sequence.org |

enhanced in resistance to lead stress, compared to the chlorotic and wilted phenotype in the control line (figure 7). In contrast with the results from lead stress, the transgenic plants did not show resistance to salt stress (electronic supplementary material, figure S5), implying *BrRBOH13* might be involved in conferring tolerance with specificity to certain abiotic conditions. Thus, it is likely to use the gene *BrRBOH13* for the potential crop improvement. And, involvement of RBOH gene family in the molecular regulation of different abiotic stresses provides a cue for future research.

In summary, an integrated analysis of phylogenetic relationships, gene structures, protein motifs, collinearity and transcriptional expression patterns on *BrRBOHs* was performed. Eight out of 14 BrRBOHs have all of four characteristic motifs of RBOH family, whereas the remaining ones just possess one to three of the motifs, indicating a great divergence during the BrRBOH evolution. With respect to their evolutionary relationships, it is likely that WGD and purifying selection were involved in the evolution of the *BrRBOHs* in *B. rapa*. The *BrRBOHs* are active not only across plant growth and development, but also in association with environmental resilience, both of which are coordinated with the structural and functional diversification of *RBOHs* during evolution.

# 4. Material and methods

## 4.1. *RBOH* gene identification

NOX family members were identified from 13 sequenced genomes (table 3). Hidden Markov models (HMMs) for Pfam IDs of RBOH proteins, i.e. NADPH_Ox (PF08414), Ferric_reduct (PF01794), FAD_binding_8 (PF08022) and NAD_binding_6 (PF08030), were retrieved from the Pfam database (http://pfam.xfam.org/), using the *Arabidopsis* RBOH protein sequence (AtrbohA, Genbank No. NP_196356.1). The HMMs were served as queries for scanning plant genome databases by the BLASTP program ($E$-value $= 1 \times 10^{-10}$). Owing to blasting against the protein sequences from annotated genomes, both the complete and incomplete annotations have been considered. And, the sequences without any annotations were filtered out from the retrieval. After validation by the online SMART tool (http://smart.embl-heidelberg.de/smart/batch.pl), the identified sequences were filtered by multiple-sequence alignment using the software MAFFT in MEGA7 to remove sequences of different transcripts from one gene.

## 4.2. Phylogenetic tree construction

According to their amino acid sequences, the candidate RBOHs were used to construct a phylogenetic tree with 1000 bootstrap replications through the NJ method in MEGA7. For the phylogenetic tree built by

RAxML (Randomized Axelerated Maximum Likelihood), the online tool (https://raxml-ng.vital-it.ch) was applied.

## 4.3. Analysis of *RBOH* gene structures, protein motifs and functional divergence

The gene structures of the *RBOH*s were analysed via the GSDS server (http://gsds.cbi.pku.edu.cn/) using genomic and protein-coding sequence information obtained from the aforementioned genome databases. Motif analysis of the RBOHs was carried out using both the MEME (http://meme-suite.org/tools/meme) and PIECE programs (http://aegilops.wheat.ucdavis.edu/piece) with default parameters. The theoretical isoelectric point (IP) and molecular mass of each RBOH protein was predicted using the ExPASy online service (http://www.expasy.ch/tools). Functional divergence, including types I and II, of the amino acid sequences of the RBOHs was analysed using the DIVERGE program [24–26]. On the basis of previous reports [27], amino acid substitutions with a posterior probability of $Q_K > 0.9$ were considered pivotal ones leading to functional divergence.

## 4.4. Chromosomal localization and collinearity analysis of *RBOH* genes

According to the genome data, information on the chromosomal localization of individual *RBOH* genes was collected and visualized using MapInspect (https://mapinspect.software.informer.com). Collinearity analysis was carried out using the program MCScanX [28]. Patterns of gene duplication were determined according to previous methods [29,30]. Briefly, genes that were evolutionarily related and had a tandem distribution on one chromosome were considered to be tandemly duplicated [30]. To determine segmental duplicates, the *RBOH*s were set as anchor sites based on their chromosomal locations; then, the protein-coding genes in flanking positions both up- and downstream of each *NOX* were compared. The criterion for defining synteny blocks is the identification of three or more conserved homologous genes within 100 kb between genomes (BlastP *E*-value $< 10^{-10}$). *Ka* (non-synonymous nucleotide substitutions) to *Ks* (synonymous nucleotide substitutions) ratios were calculated using the software DNASP5.

## 4.5. Quantitative analysis of *RBOH* expression in *Arabidopsis* and *B. rapa*

The expression patterns of *AtRBOH*s and *BrRBOH*s in *Arabidopsis* and *B. rapa* were assayed using qRT–PCR, respectively. With the specific primers (electronic supplementary material, table S3), PCR conditions were set as follows: initial denaturation of 4 min at 94°C and then 25 cycles of 30 s at 94°C, 40 s at 65°C and 1 min at 72°C with a final 5 min extension at 72°C. Relative expression values were estimated using *tubulin* as an internal control. Each reaction was performed in triplicate.

## 4.6. Transgenic *Arabidopsis* overexpressing *BrRBOH13*

*BrRBOH13* was cloned and constructed into the plasmid pCAMBIA1304 regulated by the *CAM35S* promoter. Subsequently, the recombinant *35S::BrRBOH13* was transformed into the wild-type *Arabidopsis* (Col-0) using the floral dip method. Histochemical staining for GUS activity in plants was performed according to the previous method [31]. For lead stress, $Pb(NO_3)_2 \cdot 4H_2O$ solution (0.05 mM) was irrigated into the pots cultivated with three-week *Arabidopsis* seedlings for 2 or 7 days, respectively. For salt stress, the seedlings were irrigated with 200 mM NaCl for 7 days.

Data accessibility. Websites for genome data in this study have been listed in table 3. Excel spreadsheet with list of all identified RBOH genes in the present study, containing information on gene name, chromosomal distribution, physical position, phylogenetic subgroup, numbers of exon, intron, untranslated region (UTR), peptide length, molecular mass and IP, has been uploaded as the electronic supplementary material, together with other experimental data.
Authors' contributions. D.L. conceived of the study, and designed the study; D.L., D.W., S.L., Y.D. and Y.C. carried out the experimental work and data analysis; D.L. wrote the manuscript. All authors gave final approval for publication.
Competing interests. We declare that we have no competing interests.
Funding. The research was supported by the grants from the Key University Science Research Project of Anhui Province, China (no. KJ2016A225), the Natural Science Foundation of Anhui Province, China (no. 1808085MC58), the Scientific Research Foundation for the Returned Overseas Chinese Scholars, State Education Ministry, China (no. 2015-1098), the Provincial Quality Engineer Fund of Anhui Education Department, China (no. 2015GXK015) and the State Key Laboratory of Cotton Biology Open Fund, China (no. CB2018A13).

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
