## [Reviewer comments · Royal Society Open Science]

Review History

RSOS-181727.R0 (Original submission)

Review form: Reviewer 1

Is the manuscript scientifically sound in its present form?

Yes

Are the interpretations and conclusions justified by the results?

Yes

Is the language acceptable?

Yes

Is it clear how to access all supporting data?

No

Do you have any ethical concerns with this paper?

No

Have you any concerns about statistical analyses in this paper?

No

Recommendation?

Accept with minor revision (please list in comments)

Comments to the Author(s)

The manuscript by Wu et al performs a detailed and comprehensive characterization of the evolution of the NAD oxidase family. The analysis seems highly appropriately carried out and well argued and will be of interest to researcher interested in plant stress biology. It is also well written. The exception being that I cannot really follow the logic or the scientific information regarding the transgenics. I feel that this needs to be substantially developed as it is it looks like an "add on" with little value for the rest of the manuscript having created the transgenics it seems a shame not to expose them to a range of stresses and characterize the consequences. I would recommend that the authors do so - even negative results would be interesting in this context as they would imply specificity of the results that they already show.

Review form: Reviewer 2

Is the manuscript scientifically sound in its present form?

No

Are the interpretations and conclusions justified by the results?

No

Is the language acceptable?

Yes

Is it clear how to access all supporting data?

Yes

Do you have any ethical concerns with this paper?

No

Have you any concerns about statistical analyses in this paper?

No

Recommendation?

Reject

Comments to the Author(s)

In plants, RBOHs, aka NADPH oxidases (NOXs), catalyze the biosynthesis of ROS, which are involved in a variety of physiological processes. In this manuscript, the authors identified 134 RBOH homologs in 13 plant genomes. Through phylogenetic analysis, these RBOH genes were divided into 7 subgroups. The structures and motif distributions were analyzed in these RBOH genes. Specifically, in *Brassica rapa*, the authors proposed that segmental duplication was the main cause of RBOH gene family expansion in *B. rapa*, based on their genomic collinearity analysis. In addition, the expression of RBOHs in *B. rapa* and *Arabidopsis* was analyzed. Lastly,

Arabidopsis thaliana transgenic lines overexpressing BrRBOH13 exhibited enhanced tolerance to the heavy metal lead, compared with the wild-type plants.

Overall, this manuscript provides a rather complete analysis on the phylogeny of RBOHs in a few angiosperms, a moss, and a lycophyte with detailed information of RBOHs *B. rapa*, and the putative function of BrRBOH13 in controlling plant tolerance to lead was studied in transgenic *Arabidopsis* overexpressing BrRBOH13.

Major comments:

Firstly, a very important part of the manuscript is the comprehensive RBOH gene identification in 13 plant genomes and their phylogenetic relationships. Similar work has been published recently, for example Wang, et al. ("Characterization of Rice NADPH Oxidase Genes and Their Expression under Various Environmental Conditions." *International Journal of Molecular Sciences* (2013) 14(5): 9440-9458.) and Zhang, et al. ("Genomic, molecular evolution, and expression analysis of NOX genes in soybean (*Glycine max*)." *Genomics* (2018)). In addition to identifying the RBOHs in the species of interest, these papers also identified the RBOH genes in multiple plant genomes and analyzed their phylogenetic relationships. These papers and this manuscript have many species in common, and the results should be compared. Particularly, the division of subgroups and the gene numbers are inconsistent with these previous works.

Secondly, "segmental duplication rather than tandem duplication and having played an important role in BrRBOH gene expansion", which is repeatedly emphasized by the author as an important result. It can be seen from the discussion of P5-L23 to L27 and P11-L11 to L16 and the display of figure 5 that the authors' approach to determine segmental duplication only considered whether the RBOH gene and the flanking genes are in synteny blocks. But because of the widespread presence of whole genome duplication (WGD) and recombination, this method likely finds synteny blocks generated by WGD, which are fundamentally different from the segmental duplication emphasized by the authors, and the method used in this manuscript cannot distinguish between the segmental duplication and WGD. From the results in Figure 5, the authors compared RBOH in *B. rapa* with homologs in *Arabidopsis*, according to Wang, X., et al. ("The genome of the mesopolyploid crop species *Brassica rapa*." *Nat Genet* (2011) 43(10): 1035-1039.), *B. rapa* genome has a complex history in Brassicaceae, including a unique whole genome triplication (WGT) event. The AtRBOH07(Chr05) and BrRBOH05(Chr03) BrRBOH14 (Chr10) gene pairs used by the authors to support segmental duplication in the Figure 5 are just on the synteny blocks of WGT that is supported by Wang et al (2011) (Figure 3). Therefore, segmental duplication should not be emphasized unless it is supported by more stringent evidence.

Minor comments:

P3-L44 and P10-L29: Why are there redundant sequences, is it possible that the genome has recently doubled?

P3-L53: When examining the contraction and expansion of gene families, it is important to provide a statistical support for the evolutionary inferences. CAFE or BadiRates can be used.

P4-L23 to L29 and Figure 2: The 134 sequences in the phylogenetic trees are all RBOH sequences identified by the authors, without any outgroup sequences. However, the phylogenetic tree in Fig. 4 is misleadingly shown as a rooted tree.

P6-L9: The subgroup VII mentioned here seems to be a mistake, but actually is subgroup III.

P7-L40: Most portion of a plant genome is non-coding, so the number of genes and the size of the genome are often uncorrelated. This comparison here is not really appropriate.

P10-L23: The method for identification of RBOH genes is not clear. Firstly, the specific Arabidopsis gene IDs and the Pfam IDs should be provided. The software and parameters used for the scanning should be provided. were only the annotated protein sequences scanned? If so, consider whether the genome annotation could be incomplete. Furthermore, in addition to the required motifs, whether the motifs should be arranged in a certain order should also be considered. For example, in Fig. 4, PpRBOH02 and the genes in the subgroup VII is hard to believe that they are RBOH genes.

P10-L34: The amount of data is not large. I would recommend using RaxML to build the tree, which could be more reliable.

Figure 1. The centromeres should be drawn.

Figure 4. The different colors represent different motifs. However, the information of the corresponding motifs is not indicated in the figure legend or in the figure. Furthermore, a supplemental figure can be added, in which the alignment of all proteins is provided, and boxes with different colors can be used to indicate different motifs.

Figure 6. The color scheme of the heatmap is counterintuitive. It would be better to use warm to cold colors to represent high and low expression levels.

Figure 7. It is unclear why Arabidopsis plants overexpressing BrRBOH13 also showed strong GUS activity. Was the GUS gene fused with the promoter of BrRBOH13? If so, why did leaves also show GUS activity, given that BrRBOH13 is root specific. Has the function of the homolog of BrRBOH13 in Arabidopsis been studied?

Decision letter (RSOS-181727.R0)

30-Nov-2018

Dear Dr Li,

The editors assigned to your paper ("Evolutionary and functional analysis of the plant-specific NADPH oxidase gene family in Brassica rapa L.") have now received comments from reviewers. We would like you to revise your paper in accordance with the referee and Associate Editor suggestions which can be found below (not including confidential reports to the Editor). Please note this decision does not guarantee eventual acceptance.

Please submit a copy of your revised paper before 23-Dec-2018. Please note that the revision deadline will expire at 00.00am on this date. If we do not hear from you within this time then it will be assumed that the paper has been withdrawn. In exceptional circumstances, extensions may be possible if agreed with the Editorial Office in advance. We do not allow multiple rounds of revision so we urge you to make every effort to fully address all of the comments at this stage. If deemed necessary by the Editors, your manuscript will be sent back to one or more of the original reviewers for assessment. If the original reviewers are not available, we may invite new reviewers.

To revise your manuscript, log into <http://mc.manuscriptcentral.com/rsos> and enter your Author Centre, where you will find your manuscript title listed under "Manuscripts with

Decisions." Under "Actions," click on "Create a Revision." Your manuscript number has been appended to denote a revision. Revise your manuscript and upload a new version through your Author Centre.

- Data accessibility

If you wish to submit your supporting data or code to Dryad (<http://datadryad.org/>), or modify your current submission to dryad, please use the following link:
<http://datadryad.org/submit?journalID=RSOS&manu=RSOS-181727>

- Competing interests

- Authors' contributions

AB carried out the molecular lab work, participated in data analysis, carried out sequence alignments, participated in the design of the study and drafted the manuscript; CD carried out the statistical analyses; EF collected field data; GH conceived of the study, designed the study,

coordinated the study and helped draft the manuscript. All authors gave final approval for publication.

- Acknowledgements

- Funding statement

Please note that Royal Society Open Science charge article processing charges for all new submissions that are accepted for publication. Charges will also apply to papers transferred to Royal Society Open Science from other Royal Society Publishing journals, as well as papers submitted as part of our collaboration with the Royal Society of Chemistry (<http://rsos.royalsocietypublishing.org/chemistry>). If your manuscript is newly submitted and subsequently accepted for publication, you will be asked to pay the article processing charge, unless you request a waiver and this is approved by Royal Society Publishing. You can find out more about the charges at <http://rsos.royalsocietypublishing.org/page/charges>. Should you have any queries, please contact openscience@royalsociety.org.

on behalf of Professor Kevin Padian (Subject Editor)
openscience@royalsociety.org

Subject Editor's comments:

It is important to establish the novelty of this work, in accordance with the comments of the second reviewer; and please see other comments that need to be addressed carefully. Best of luck with your revision.

Associate Editor's comments:

The second referee in particular has substantial feedback for the authors, and we encourage them to respond as best they can to this commentary - ideally by incorporating these suggestions in the manuscript itself or, if this is not practical, to provide a reasoned scientific rebuttal to the reviewer. The journal's policy generally precludes multiple rounds of revision, and if the reviewer remains unsatisfied after revision, it may not be possible to further consider the revision. Good luck!

Comments to Author:

Reviewers' Comments to Author:
Reviewer: 1

Comments to the Author(s)

The manuscript by Wu et al performs a detailed and comprehensive characterization of the evolution of the NAD oxidase family. The analysis seems highly appropriately carried out and

well argued and will be of interest to researcher interested in plant stress biology. It is also well written. The exception being that I cannot really follow the logic or the scientific information regarding the transgenics. I feel that this needs to be substantially developed as it is it looks like an "add on" with little value for the rest of the manuscript having created the transgenics it seems a shame not to expose them to a range of stresses and characterize the consequences. I would recommend that the authors do so - even negative results would be interesting in this context as they would imply specificity of the results that they already show.

Reviewer: 2

Comments to the Author(s)

In plants, RBOHs, aka NADPH oxidases (NOXs), catalyze the biosynthesis of ROS, which are involved in a variety of physiological processes. In this manuscript, the authors identified 134 RBOH homologs in 13 plant genomes. Through phylogenetic analysis, these RBOH genes were divided into 7 subgroups. The structures and motif distributions were analyzed in these RBOH genes. Specifically, in *Brassica rapa*, the authors proposed that segmental duplication was the main cause of RBOH gene family expansion in *B. rapa*, based on their genomic collinearity analysis. In addition, the expression of RBOHs in *B. rapa* and *Arabidopsis* was analyzed. Lastly, *Arabidopsis thaliana* transgenic lines overexpressing BrRBOH13 exhibited enhanced tolerance to the heavy metal lead, compared with the wild-type plants.

Overall, this manuscript provides a rather complete analysis on the phylogeny of RBOHs in a few angiosperms, a moss, and a lycophyte with detailed information of RBOHs *B. rapa*, and the putative function of BrRBOH13 in controlling plant tolerance to lead was studied in transgenic *Arabidopsis* overexpressing BrRBOH13.

Major comments:

Firstly, a very important part of the manuscript is the comprehensive RBOH gene identification in 13 plant genomes and their phylogenetic relationships. Similar work has been published recently, for example Wang, et al. ("Characterization of Rice NADPH Oxidase Genes and Their Expression under Various Environmental Conditions." *International Journal of Molecular Sciences* (2013) 14(5): 9440-9458.) and Zhang, et al. ("Genomic, molecular evolution, and expression analysis of NOX genes in soybean (*Glycine max*)." *Genomics* (2018)). In addition to identifying the RBOHs in the species of interest, these papers also identified the RBOH genes in multiple plant genomes and analyzed their phylogenetic relationships. These papers and this manuscript have many species in common, and the results should be compared. Particularly, the division of subgroups and the gene numbers are inconsistent with these previous works.

Secondly, "segmental duplication rather than tandem duplication and having played an important role in BrRBOH gene expansion", which is repeatedly emphasized by the author as an important result. It can be seen from the discussion of P5-L23 to L27 and P11-L11 to L16 and the display of figure 5 that the authors' approach to determine segmental duplication only considered whether the RBOH gene and the flanking genes are in synteny blocks. But because of the widespread presence of whole genome duplication (WGD) and recombination, this method likely finds synteny blocks generated by WGD, which are fundamentally different from the segmental duplication emphasized by the authors, and the method used in this manuscript cannot distinguish between the segmental duplication and WGD. From the results in Figure 5, the authors compared RBOH in *B. rapa* with homologs in *Arabidopsis*, according to Wang, X., et al. ("The genome of the mesopolyploid crop species *Brassica rapa*." *Nat Genet* (2011) 43(10): 1035-1039.), *B. rapa* genome has a complex history in Brassicaceae, including a unique whole genome triplication (WGT) event. The AtRBOH07(Chr05) and BrRBOH05(Chr03) BrRBOH14 (Chr10) gene pairs used by the authors to support segmental duplication in the Figure 5 are just on the synteny

blocks of WGT that is supported by Wang et al (2011) (Figure 3). Therefore, segmental duplication should not be emphasized unless it is supported by more stringent evidence.

Minor comments:

P3-L44 and P10-L29: Why are there redundant sequences, is it possible that the genome has recently doubled?

P3-L53: When examining the contraction and expansion of gene families, it is important to provide a statistical support for the evolutionary inferences. CAFE or BadiRates can be used.

P4-L23 to L29 and Figure 2: The 134 sequences in the phylogenetic trees are all RBOH sequences identified by the authors, without any outgroup sequences. However, the phylogenetic tree in Fig. 4 is misleadingly shown as a rooted tree.

P6-L9: The subgroup VII mentioned here seems to be a mistake, but actually is subgroup III.

P7-L40: Most portion of a plant genome is non-coding, so the number of genes and the size of the genome are often uncorrelated. This comparison here is not really appropriate.

P10-L23: The method for identification of RBOH genes is not clear. Firstly, the specific Arabidopsis gene IDs and the Pfam IDs should be provided. The software and parameters used for the scanning should be provided. were only the annotated protein sequences scanned? If so, consider whether the genome annotation could be incomplete. Furthermore, in addition to the required motifs, whether the motifs should be arranged in a certain order should also be considered. For example, in Fig. 4, PpRBOH02 and the genes in the subgroup VII is hard to believe that they are RBOH genes.

P10-L34: The amount of data is not large. I would recommend using RaxML to build the tree, which could be more reliable.

Figure 1. The centromeres should be drawn.

Figure 4. The different colors represent different motifs. However, the information of the corresponding motifs is not indicated in the figure legend or in the figure. Furthermore, a supplemental figure can be added, in which the alignment of all proteins is provided, and boxes with different colors can be used to indicate different motifs.

Figure 6. The color scheme of the heatmap is counterintuitive. It would be better to use warm to cold colors to represent high and low expression levels.

Figure 7. It is unclear why Arabidopsis plants overexpressing BrRBOH13 also showed strong GUS activity. Was the GUS gene fused with the promoter of BrRBOH13? If so, why did leaves also show GUS activity, given that BrRBOH13 is root specific. Has the function of the homolog of BrRBOH13 in Arabidopsis been studied?

Author's Response to Decision Letter for (RSOS-181727.R0)

See Appendix A.

RSOS-181727.R1 (Revision)

Review form: Reviewer 1

Is the manuscript scientifically sound in its present form?

Yes

Are the interpretations and conclusions justified by the results?

Yes

Is the language acceptable?

Yes

Is it clear how to access all supporting data?

Yes

Do you have any ethical concerns with this paper?

No

Have you any concerns about statistical analyses in this paper?

No

Recommendation?

Accept as is

Comments to the Author(s)

The revised manuscript has greatly improved its strongest part and added additional information concerning the transgenics that they created. While I still find this section relatively superficial and the manuscript slightly unbalanced I appreciate the authors efforts and thus recommend that the manuscript is published

Review form: Reviewer 2

Is the manuscript scientifically sound in its present form?

No

Are the interpretations and conclusions justified by the results?

Yes

Is the language acceptable?

Yes

Is it clear how to access all supporting data?

Yes

Do you have any ethical concerns with this paper?

No

Have you any concerns about statistical analyses in this paper?

No

Recommendation?

Major revision is needed (please make suggestions in comments)

Comments to the Author(s)

The revised manuscript has undergone a major revision and addressed most of my concerns. Please find my comments below.

1. P 9, lines 29-39. Although the author mentioned the function of whole genome duplication (WGD) in the expansion of RBOH gene family in the Abstract and Discussion, no analysis was done to determine which BrRBOH genes were duplicated by WGD and which originated from segmental duplications. As recommended in the previous review, data from Wang et al. ("The genome of the mesopolyploid crop species *Brassica rapa*. "Nat Genet, 2011, 43: 1035-1039) can be used to determine which BrRBOH genes originated from WGD, and which BrRBOHs were from segmental duplications. Here, it is probably necessary to stress again that segmental duplication is not the same as WGD, and some of the *B. rapa* genes are from WGD but not segmental duplications, and this is why it is necessary to distinguish these two modes of duplications in this work.

2. P6, lines 13-25. The genes in subgroup VII lack some motifs and have large variations from genes in the other subgroups, and they were used as an outgroup in the entire phylogenetic tree. This approach is quite debatable, because lacking motifs and accumulated mutations in the gene sequences could be caused by pseudogenization or annotation errors. The selection of outgroup genes should be from another gene family which is very close to RBOHs. If there are no suitable genes as an outgroup, I would recommend using a unroot tree instead.

3. The author mentioned in the Method that BLASTP was used as the scanning tool (P32, line 39), then the author must have blasted against the protein sequences from annotated genomes. If so, the impact of incomplete annotations should be considered. In P29, line 56-57, "When the identification of RBOH members was carried out, all genome sequences have been scanned whether they have annotation or not", indicating that the author had considered this problem, but how this was done should be described in the Methods as well.

Decision letter (RSOS-181727.R1)

23-Jan-2019

Dear Dr Li:

On behalf of the Editors, I am pleased to inform you that your Manuscript RSOS-181727.R1 entitled "Evolutionary and functional analysis of the plant-specific NADPH oxidase gene family in *Brassica rapa* L." has been accepted for publication in Royal Society Open Science subject to minor revision in accordance with the referee suggestions. Please find the referees' comments at the end of this email.

The reviewers and Subject Editor have recommended publication, but also suggest some minor revisions to your manuscript. Therefore, I invite you to respond to the comments and revise your manuscript.

- Ethics statement

- Data accessibility

<http://datadryad.org/submit?journalID=RSOS&manu=RSOS-181727.R1>

- Competing interests

- Authors' contributions

- Acknowledgements

- Funding statement

Please note that we cannot publish your manuscript without these end statements included. We have included a screenshot example of the end statements for reference. If you feel that a given

heading is not relevant to your paper, please nevertheless include the heading and explicitly state that it is not relevant to your work.

Because the schedule for publication is very tight, it is a condition of publication that you submit the revised version of your manuscript before 01-Feb-2019. Please note that the revision deadline will expire at 00.00am on this date. If you do not think you will be able to meet this date please let me know immediately.

on behalf of Professor Kevin Padian (Subject Editor)
openscience@royalsociety.org

Associate Editor Comments to Author:

The reviewers of your paper are broadly in favour of publication, though you'll note that one of the referees has a number of remaining queries you must address before the journal will accept the paper. Please ensure you incorporate the changes the referee requests -- it will not be possible to publish the paper without these changes included as requested by the referee.

Reviewer comments to Author:

Reviewer: 2

Comments to the Author(s)

The revised manuscript has undergone a major revision and addressed most of my concerns. Please find my comments below.

1. P 9, lines 29-39. Although the author mentioned the function of whole genome duplication (WGD) in the expansion of RBOH gene family in the Abstract and Discussion, no analysis was done to determine which BrRBOH genes were duplicated by WGD and which originated from segmental duplications. As recommended in the previous review, data from Wang et al. ("The genome of the mesopolyploid crop species *Brassica rapa*. "Nat Genet, 2011, 43: 1035-1039) can be used to determine which BrRBOH genes originated from WGD, and which BrRBOHs were from segmental duplications. Here, it is probably necessary to stress again that segmental duplication is not the same as WGD, and some of the *B. rapa* genes are from WGD but not segmental duplications, and this is why it is necessary to distinguish these two modes of duplications in this work.

2. P6, lines 13-25. The genes in subgroup VII lack some motifs and have large variations from genes in the other subgroups, and they were used as an outgroup in the entire phylogenetic tree. This approach is quite debatable, because lacking motifs and accumulated mutations in the gene sequences could be caused by pseudogenization or annotation errors. The selection of outgroup genes should be from another gene family which is very close to RBOHs. If there are no suitable genes as an outgroup, I would recommend using a unroot tree instead.

3. The author mentioned in the Method that BLASTP was used as the scanning tool (P32, line 39), then the author must have blasted against the protein sequences from annotated genomes. If so, the impact of incomplete annotations should be considered. In P29, line 56-57, "When the identification of RBOH members was carried out, all genome sequences have been scanned whether they have annotation or not", indicating that the author had considered this problem, but how this was done should be described in the Methods as well.

Reviewer: 1

Comments to the Author(s)

The revised manuscript has greatly improved its strongest part and added additional information concerning the transgenics that they created. While I still find this section relatively superficial and the manuscript slightly unbalanced I appreciate the authors efforts and thus recommend that the manuscript is published

Author's Response to Decision Letter for (RSOS-181727.R1)

See Appendix B.

Decision letter (RSOS-181727.R2)

31-Jan-2019

Dear Dr Li,

I am pleased to inform you that your manuscript entitled "Evolutionary and functional analysis of the plant-specific NADPH oxidase gene family in *Brassica rapa* L." is now accepted for publication in Royal Society Open Science.

on behalf of Professor Kevin Padian (Subject Editor)
openscience@royalsociety.org

Appendix A

Royal Society Open Science - Manuscript ID RSOS-181727

Evolutionary and functional analysis of the plant-specific NADPH oxidase gene family in *Brassica rapa* L.

Dahui Li, Di Wu, Shizhou Li, Yu Dai, and Yunpeng Cao

Dear Editor:

Thank you for your critical reading of our manuscript (RSOS-181727). And we have revised the manuscript in accordance with the reviewers' comments. Enclosed is a revised version of our manuscript, where the changes are highlighted in blue. Moreover, figure 1, figure 2, figure 6, and figure S1 are re-prepared. Previous figures 3 and 4 are integrated into one figure (i.e., figure 4). Figure 3, and two electronic supplemental files (i.e., figures S2 and S5) are added into the revised manuscript.

The questions of the reviewers and our answers are listed in the following:

To Reviewer #1:

1. [The manuscript by Wu et al performs a detailed and comprehensive characterization of the evolution of the NAD oxidase family. The analysis seems highly appropriately carried out and well argued and will be of interest to researcher interested in plant stress biology. It is also well written. The exception being that I cannot really follow the logic or the scientific information regarding the transgenics. I feel that this needs to be substantially developed as it is it looks like an "add on" with little value for the rest of the manuscript having created the transgenics it seems a shame not to expose them to a range of stresses and characterize the consequences. I would recommend that the authors do so - even negative results would be interesting in this context as they would imply specificity of the results that they already show.]

[Thank you for your valuable suggestion. In this study, *BrRBOH13* transgenic Arabidopsis have been treated with both abiotic stresses of heavy metal lead and salt NaCl, respectively. In contrast to the results from lead stress, the transgenic plants did not showed

resistance to salt stress. We have revised this content and supplemented as an electronic file, figure S5, presenting its phenotype under salt stress, in the revised manuscript.

(1) (Results section, page 7, line 24) These *BrRBOH13* transgenic lines treated with both abiotic stresses of heavy metal lead and salt NaCl, respectively. As showed in figure 7D–7G, the transgenic *Arabidopsis* plants overexpressing *BrRBOH13* have increased tolerance to heavy metal lead (figure 7D – 7G). Upon 0.05 mM Pb^{2+} -treatments, the *BrRBOH13* transgenic plants were able to keep normal growth throughout the duration from 2 to 7 days (figure 7D and 7E), whereas wild type *Arabidopsis* has gradually undergone chlorosis and turned into senescent phenotype (figure 7F and 7G). After the treatment with salt stress (200 mM NaCl) for 7 days, the transgenic plants showed the appearance of both chlorosis and necrotic lesions on their leaves, compared to the control ones without NaCl-treatment (electronic supplementary material, figure S4). This result suggested that no enhanced salt tolerance was activated in *BrRBOH13* transgenic *Arabidopsis* plants.

(2) (Discussion section, page 10, line 27) In the present research, the *BrRBOH13* transgenic *Arabidopsis* plants were enhanced in resistance to lead stress, compared to the chlorotic and wilted phenotype in the control line (figure 7). In contrast to the results from lead stress, the transgenic plants did not showed resistance to salt stress (electronic supplementary material, figure S4), implying *BrRBOH13* might be involved in conferring tolerance with specificity to certain abiotic conditions. Thus, it is likely to utilize the gene *BrRBOH13* for the potential crop improvement. And involvement of RBOH gene family in the molecular regulation of different abiotic stresses, provides a cue for future research.

(3) (Material and methods section, page 13, line 2) For salt stress, the seedlings were irrigated with 200 mM NaCl for 7 days.]

To Reviewer #2:

1. [Firstly, a very important part of the manuscript is the comprehensive RBOH gene identification in 13 plant genomes and their phylogenetic relationships. Similar work has been published recently, for example Wang, et al. ("Characterization of Rice NADPH Oxidase Genes and Their Expression under Various

Environmental Conditions." *International Journal of Molecular Sciences* (2013) 14(5): 9440-9458.) and Zhang, et al. ("Genomic, molecular evolution, and expression analysis of NOX genes in soybean (*Glycine max*)." *Genomics* (2018)). In addition to identifying the RBOHs in the species of interest, these papers also identified the RBOH genes in multiple plant genomes and analyzed their phylogenetic relationships. These papers and this manuscript have many species in common, and the results should be compared. Particularly, the division of subgroups and the gene numbers are inconsistent with these previous works.]

[Thank you for your valuable suggestion. We have revised this part of text in accordance with your comments.

(1) Two of the published papers have been added into the reference list: 17. Wang GF, Li WQ, Li WY, Wu GL, Zhou CY, Chen KM. 2013 Characterization of rice NADPH oxidase genes and their expression under various environmental conditions. *Int. J. Mol. Sci.* 14, 9 440–9 458. (doi:10.3390/ijms14059440); 18. Zhang ZB, Zhao YL, Feng XB, Luo ZY, Kong SW, Zhang C, Gong AD, Yuan HY, Cheng L, Wang XN. 2018 Genomic, molecular evolution, and expression analysis of NOX genes in soybean (*Glycine max*). *Genomics*, available online. (doi:10.1016/j.ygeno.2018.03.018).

(2) (Introduction section, page 3, line 3) In plants, RBOH genes and their functions have been extensively characterized in *Arabidopsis* (*Arabidopsis thaliana*), rice (*Oryza sativa*), and soybean (*Glycine max*) [4,11,17,18].

(3) (Discussion section, page 8, line 12) And the numbers of RBOHs in *Arabidopsis*, *P. trichocarpa*, rice, and soybean, were consistent with those in the previous reports [4,17,18]. Inconsistency in gene numbers was found out within four species, i.e., *Z. mays* (15), *S. bicolor* (9), *P. patens* (5), and *S. moellendorffii* (8) in the presented study, which were 12, 7, 4 and 5 in the previous work, respectively [17]. After rescreening of the respective genome data of these four species, all of RBOHs in the present study have been validated with the presence of the conserved RBOH domains (electronic supplementary material, figure S1 and figure 2).

(4) (Discussion section, page 8, line 29) In addition, compared with the previous analysis on RBOHs among soybean or rice and other

plants [17,18], some similar phylogenetic relationships were obtained. For example, four GmRBOHs (02, 16, 17, and 12) were not only formed into two paralogous pairs (GmRBOH02-GmRBOH16, and GmRBOH17-GmRBOH12), but also grouped with AtRBOH01 and three OsRBOHs (01, 09 and 08) into the division of subgroup I, while another four GmRBOHs (03, 07, 09, and 05), producing two paralogous pairs (GmRBOH03-GmRBOH07, and GmRBOH09-GmRBOH05), were grouped with four AtRBOHs (06, 09, 07, 08) into the division of subgroup II in the present study (figure 2).

(5) In the published paper by Wang, et al. ("Characterization of Rice NADPH Oxidase Genes and Their Expression under Various Environmental Conditions." International Journal of Molecular Sciences (2013) 14(5): 9440-9458.), a total of 65 RBOH proteins were recognized in nine species, including 9 in rice, 10 in Arabidopsis, 10 in *Populus trichocarpa*, 7 in *Vitis vinifera*, 1 in *Picea sitchensis*, 12 in *Zea mays*, 7 in *Sorghum bicolor*, 4 in *Physcomitrella patens*, and 5 in *Selaginella moellendorffii*. Seven out of nine species are also included in the present study. And the identified RBOHs from these seven species consisted of 9 in rice, 10 in Arabidopsis, 10 in *P. trichocarpa*, 15 in *Z. mays*, 9 in *S. bicolor*, 5 in *P. patens*, and 8 in *S. moellendorffii*. Therefore, gene numbers that are inconsistent with the previous work are presented in four species, i.e., *Z. mays* (15), *S. bicolor* (9), *P. patens* (5), and *S. moellendorffii* (8). After rescreening of the respective genome data of these four species, all of RBOHs in the present study have been validated with the presence of the conserved RBOH domains. The inconsistency in the division of subgroups on the phylogenetic trees, were likely due to the difference in the selected species for analysis, albeit 7 or 3 out of 13 species in this study were same with the previous ones, respectively [17,18].]

2. [Secondly, “segmental duplication rather than tandem duplication and having played an important role in BrRBOH gene expansion”, which is repeatedly emphasized by the author as an important result. It can be seen from the discussion of P5-L23 to L27 and P11-L11 to L16 and the display of figure 5 that the authors’ approach to determine segmental duplication only considered whether the RBOH gene and the flanking genes are in synteny blocks. But

because of the widespread presence of whole genome duplication (WGD) and recombination, this method likely finds synteny blocks generated by WGD, which are fundamentally different from the segmental duplication emphasized by the authors, and the method used in this manuscript cannot distinguish between the segmental duplication and WGD. From the results in Figure 5, the authors compared RBOH in *B. rapa* with homologs in Arabidopsis, according to Wang, X., et al. ("The genome of the mesopolyploid crop species *Brassica rapa*. "Nat Genet (2011) 43(10): 1035-1039.), *B. rapa* genome has a complex history in Brassicaceae, including a unique whole genome triplication (WGT) event. The *AtRBOH07*(Chr05) and *BrRBOH05*(Chr03) *BrRBOH14* (Chr10) gene pairs used by the authors to support segmental duplication in the Figure 5 are just on the synteny blocks of WGT that is supported by Wang et al (2011) (Figure 3). Therefore, segmental duplication should not be emphasized unless it is supported by more stringent evidence.]

[Thank you for your valuable suggestion. We have revised the related description in accordance with your comments.

(1) A paper has been added into the reference list: 22. Wang XW, Wang HZ, Wang J, Sun RF, et al. 2011 The genome of the mesopolyploid crop species *Brassica rapa*. *Nat. Genet.* **43**, 1 035–1 039. (doi:10.1038/ng.919)

(2) (Abstract section, page 1, line 18) the results suggested that whole genome duplication (WGD) might have played an important role in *BrRBOH* gene expansion.

(3) (Discussion section, page 9, line 14) It has been reported that *B. rapa* genome had undergone a complex history of evolution in *Brassicaceae*, including a unique whole genome triplication (WGT) event [22]. The gene pairs demonstrated in figure 5, such as *AtRBOH07* (Chr05)-*BrRBOH05* (Chr03), and *AtRBOH07* (Chr05)-*BrRBOH14* (Chr10), are representatives on the synteny blocks of WGT according to the evolution of *B. rapa* genome [22]. These results suggest that whole genome duplication (WGD) might have played an important role in *BrRBOH* gene expansion.

(4) (Discussion section, page 11, line 8) it is likely that WGD and purifying selection were involved in the evolution of the *BrRBOHs* in *B. rapa*.]

3. [P3-L44 and P10-L29: Why are there redundant sequences, is it possible that the genome has recently doubled?]

[Redundant sequences in the text were the sequences of different transcripts from one gene. It is our fault that we improperly described these sequences. We have deleted this incorrect description in the revised manuscript.

(1) (Results section, page 3, line 21) Four of the 138 identified NOXs were removed from the initial collection because they are different transcripts from one gene.

(2) (Material and methods section, page 11, line 22) to remove sequences of different transcripts from one gene.]

4. [P3-L53: When examining the contraction and expansion of gene families, it is important to provide a statistical support for the evolutionary inferences. CAFE or BadiRates can be used.]

[Thank you for your valuable suggestion. After the analysis of RBOH gene family evolution using CAFE, no expansion or decrease in RBOH gene family was found out between moss *Physcomitrella patens* and the angiosperm species examined [with Family-wide P-value: 0.270000 and Viterbi P-values ((-, -), (-, -), (-, -), (-, -), (-, -), (-, -), (-, -), (-, -), (-, -), (-, -), (-, -), (-, -), (-, -))].

Therefore, we have revised this part of content in the revised manuscript.

(Results section, page 3, line 26) Number of the NOX family members was varied in the angiosperm species examined.]

5. [P4-L23 to L29 and Figure 2: The 134 sequences in the phylogenetic trees are all RBOH sequences identified by the authors, without any outgroup sequences. However, the phylogenetic tree in Fig. 4 is misleadingly shown as a rooted tree.]

[Thank you for your comments. It is a careless mistake when we prepared the previous both Fig. 4 and Fig. 2. The phylogenetic tree presented in the previous Fig. 4 and Fig. 2 should be an unrooted tree. We have re-prepared the figure 2 and figure 4 in the revised manuscript. The reason for your comment 'However, the

phylogenetic tree in Fig. 4 is misleadingly shown as a rooted tree.’, is likely due to a greatly divergent relationship between the subgroup VII, consisting of BrRBOH08 and 12, CsRBOH08, and ZmRBOH08, and other subgroups. In comparison with others, the members from the subgroup VII only have one or two RBOH-characteristic motifs (i.e., NADPH_Ox in BrRBOH12, CsRBOH08, and ZmRBOH08 or both NADPH_Ox and NAD_binding_6 in BrRBOH08), respectively. As a result, it was treated like an outgroup when the phylogenetic tree was constructed by the NJ method.

(Result section, page 4, line 26) Based on sequence alignment, it was found out that all of four characteristic motifs of RBOH family (NADPH_Ox, Ferric_reduct, FAD_binding_8, and NAD_binding_6) were presented within 124 out of 134 RBOHs (electronic supplementary material, figure S1 and figure 2). And there showed a high conservation of these characteristic motifs among the RBOHs identified (figure 3). Among the remaining 10 RBOHs, BrRBOH04, BrRBOH07, and BrRBOH10 have three (NADPH_Ox, Ferric_reduct, and NAD_binding_6 in RBOH04 and BrRBOH10, or NADPH_Ox, FAD_binding_8, and NAD_binding_6 in BrRBOH07), BrRBOH08, PpRBOH02 and ZmRBOH12 have two (both NADPH_Ox and NAD_binding_6 in BrRBOH08, or NADPH_Ox and Ferric_reduct in PpRBOH02 and ZmRBOH12), BrRBOH01 and BrRBOH12, CsRBOH08, and ZmRBOH08 have one (NADPH_Ox) of RBOH-characteristic motifs, respectively (figure 2). The subgroup VII, consisting of BrRBOH08 and 12, CsRBOH08, and ZmRBOH08, showed a such greatly divergent relationship with other subgroups that it was treated like an outgroup when the phylogenetic tree was constructed.]

6. [P6-L9: The subgroup VII mentioned here seems to be a mistake, but actually is subgroup III.]

[We have revised this part of content in the revised manuscript. (Result section, page 6, line 19,) To further investigate the functional divergence of amino acid sequences after RBOH duplication, a representative subgroup (III) in the phylogenetic tree was selected because it consisted of various plant lineages.

Subgroup III was classified into three clusters.

(Electronic supplementary materials section, page 23, line 16)

Figure S4. The subgroup III selected from the phylogenetic tree in the figure 2.]

7. [P7-L40: Most portion of a plant genome is non-coding, so the number of genes and the size of the genome are often uncorrelated. This comparison here is not really appropriate.]

[We have deleted this description in the revised manuscript.]

8. [P10-L23: The method for identification of RBOH genes is not clear. Firstly, the specific Arabidopsis gene IDs and the Pfam IDs should be provided. The software and parameters used for the scanning should be provided. were only the annotated protein sequences scanned? If so, consider whether the genome annotation could be incomplete. Furthermore, in addition to the required motifs, whether the motifs should be arranged in a certain order should also be considered. For example, in Fig. 4, PpRBOH02 and the genes in the subgroup VII is hard to believe that they are RBOH genes.]

[We have revised this part of content in the “Material and methods” section. (Material and methods section, page 11, line 15) Hidden Markov Models (HMMs) for Pfam IDs of RBOH proteins, i.e., NADPH_Ox (PF08414), Ferric_reduct (PF01794), FAD_binding_8 (PF08022), and NAD_binding_6 (PF08030), were retrieved from the Pfam database (<http://pfam.xfam.org/>), using the Arabidopsis RBOH protein sequence (AtrbohA, Genbank No. NP_196356.1). The HMMs were served as queries for scanning plant genome databases by the BLASTP program (E-value=1e-10).

As you have point out, it was found out that 10 out of 134 RBOHs, have one to three of RBOH-characteristic motifs, respectively (figure 2). Please refer to description in No.5 in this list. Although PpRBOH02 and the genes in the subgroup VII have two or one of RBOH-characteristic motifs, they were kept in the present study, in consideration of analysis of the evolution and divergence in RBOH family.

When the identification of RBOH members was carried out, all

genome sequences have been scanned whether they have annotation or not. In addition, the motifs arranged in a certain order have also be considered because the phylogenetic tree was constructed based on a complete alignment of full length sequences.]

9. [P10-L34: The amount of data is not large. I would recommend using RaxML to build the tree, which could be more reliable.]

[Thank you for your valuable suggestion. We have built the phylogenetic using RaxML.

(Result section, page 5, line 8) Furthermore, another phylogenetic tree was constructed using RAxML, with seven excluded from the 134 identified RBOHs, including BrRBOH01, PpRBOH02, ZmRBOH12, and four members of the subgroup VII (electronic supplementary material, figure S2). As showed in electronic supplementary material, figure S2, similar division of subgroups (I–VI) was built, compared with the previous one (figure 2).

(Material and methods section, page 11, line 25) For the phylogenetic tree built by RAxML (Randomized Axelerated Maximum Likelihood), the online tool (<https://raxml-ng.vital-it.ch>) was applied.

(Electronic supplementary materials section, page 23, line 8,) **Figure S2.** The phylogenetic tree is constructed using RAxML, after 7 members with one or two RBOH-characteristic motifs were excluded from the 134 identified RBOHs. Six subgroups are indicated with I–VI, respectively. Other symbols are the same with those in Figure 2.]

10. [Figure 1. The centromeres should be drawn.]

[Thank you for your valuable suggestion. We have revised the Figure 1 in the revised manuscript following your suggestion.

(Figure captions section, page 21, line 4) Figure 1. Genomic distribution of RBOH genes on *B. rapa* chromosomes. Chromosomal locations of RBOHs are illustrated based on the physical position of each gene. The number of chromosomes is indicated on the top of each chromosome. The centromeres are marked by red ovals, respectively. The centromere on

chromosome06 is unavailable according to the current genome data of *B. rapa*.]

11. [Figure 4. The different colors represent different motifs. However, the information of the corresponding motifs is not indicated in the figure legend or in the figure. Furthermore, a supplemental figure can be added, in which the alignment of all proteins is provided, and boxes with different colors can be used to indicate different motifs.]

[We have revised this part of content in the revised manuscript. The information of the corresponding motifs is added in the figure 4. And Sequence logos of the conserved motifs of NADPH_Ox, Ferric_reduct, FAD_binding_8, and NAD_binding_6 among 134 RBOHs are showed in the figure 3. An electronic supplementary material, figure S1, is supplied, exhibiting the alignment of all proteins.

(Electronic supplementary materials section, page 23, line 4)

Figure S1. A complete protein sequence alignment of 134 RBOHs. Four domains of NADPH_Ox, Ferric_reduct, FAD_binding_8, and NAD_binding_6 are marked by boxes in green, yellow, blue, and black, respectively.]

12. [Figure 6. The color scheme of the heatmap is counterintuitive. It would be better to use warm to cold colors to represent high and low expression levels.]

[We have revised the figure 6 following your suggestion.]

13. [Figure 7. It is unclear why Arabidopsis plants overexpressing BrRBOH13 also showed strong GUS activity. Was the GUS gene fused with the promoter of BrRBOH13? If so, why did leaves also show GUS activity, given that BrRBOH13 is root specific. Has the function of the homolog of BrRBOH13 in Arabidopsis been studied?]

[In this study, the GUS gene was not fused with the promoter of *BrRBOH13*. The plasmid pCAMBIA1304 itself has already been integrated with both the promoter of *CAM35S* and reporter gene

GUS. When *BrRBOH13* was cloned and constructed into the plasmid pCAMBIA1304, *BrRBOH13* and *GUS* gene are in a tandem form, both of which are regulated by the *CAM35S* promoter. Subsequently, the recombinant *35S::BrRBOH13* was transformed into the wild type *Arabidopsis* (Col-0). The *CAM35S* promoter is constitutive. Therefore, the gene under its regulation could be constitutively expressed. Please refer to the “Material and methods” section, page 12, line 27.]

Obediently yours,
Dahui Li

College of Life Science,
Anhui Agricultural University,
Hefei 230036, China
Telephone: +86-551-6578-6237
E-mail: dahui2@126.com

Appendix B

Royal Society Open Science - Manuscript ID RSOS-181727.R1
Evolutionary and functional analysis of the plant-specific NADPH
oxidase gene family in *Brassica rapa* L.
Dahui Li, Di Wu, Shizhou Li, Yu Dai, and Yunpeng Cao

Dear Editor:

Thank you for your critical reading of our manuscript (RSOS-181727.R1). And we have revised the manuscript in accordance with the reviewers' comments. Enclosed is a revised version of our manuscript, where the changes are highlighted in blue. Additionally, error correction in the centromere position within the figure 1, has been performed in the revised manuscript.

The questions of the reviewers and our answers are listed in the following:

To Reviewer #2:

1. [P 9, lines 29-39. Although the author mentioned the function of whole genome duplication (WGD) in the expansion of RBOH gene family in the Abstract and Discussion, no analysis was done to determine which BrRBOH genes were duplicated by WGD and which originated from segmental duplications. As recommended in the previous review, data from Wang et al. ("The genome of the mesopolyploid crop species *Brassica rapa*. "Nat Genet, 2011, 43: 1035-1039) can be used to determine which BrRBOH genes originated from WGD, and which BrRBOHs were from segmental duplications. Here, it is probably necessary to stress again that segmental duplication is not the same as WGD, and some of the *B. rapa* genes are from WGD but not segmental duplications, and this is why it is necessary to distinguish these two modes of duplications in this work.]

[Thank you for your valuable suggestion. We have revised this part of text in accordance with your comments.

(Discussion section, page 9, line 15) The gene pairs demonstrated in figure 5 and table 1, including two gene-pairs within *BrRBOHs* (*BrRBOH04/BrRBOH10*, *BrRBOH05/BrRBOH14*) and 15 gene-pairs between *BrRBOHs* and *AtRBOHs*, are representatives on the synteny blocks of WGT according to the evolution of *B. rapa* genome [22]. Three of five pairs within *BrRBOHs*, i.e., *BrRBOH01/BrRBOH06*, *BrRBOH02/BrRBOH09*, and *BrRBOH13/BrRBOH14*, were from segmental duplications (table 1).]

2. [P6, lines 13-25. The genes in subgroup VII lack some motifs and have large variations from genes in the other subgroups, and they were used as an outgroup in the entire phylogenetic tree. This approach is quite debatable, because lacking motifs and accumulated mutations in the gene sequences could be caused by pseudogenization or annotation errors. The selection of outgroup genes should be from another gene family which is very close to RBOHs. If there are no suitable genes as an outgroup, I would recommend using a unroot tree instead.]

[Thank you for your valuable suggestion. We have revised the phylogenetic tree using a unroot one (figure 2), presented previously as electronic supplementary material, figure S2, which are constructed with seven excluded from the 134 identified RBOHs, including *BrRBOH01*, *PpRBOH02*, *ZmRBOH12*, and four members of the subgroup VII.

(1) (Results section, page 5, line 5) Together with *BrRBOH01*, *PpRBOH02*, and *ZmRBOH12*, four members of the subgroup VII (*BrRBOH08*, *BrRBOH12*, *CsRBOH08*, and *ZmRBOH08*) were excluded from the 134 identified RBOHs and the construction of a unroot phylogenetic tree using RAxML (figure 2), because of their great divergence with other RBOH members. As showed in figure 2, similar division of subgroups (I–VI) was built, compared with the previous one (electronic supplementary material, figure S2).

(2) (Electronic supplementary materials section, page 23, line 8) Figure S2. The phylogenetic tree (circle a) is constructed using NJ method, according to a complete protein sequence alignment of

134 RBOHs. Seven subgroups are indicated with I–VII, respectively. Triangles in red, blue, purple, and green color indicate domains of NADPH_Ox, Ferric_reduct, FAD_binding_8, and NAD_binding_6, respectively. Circles b, c, d, and e are composed of four above mentioned domains within each RBOHs, respectively. Circles at the individual nodes represent bootstrap support.

(3) (Figure captions section, page 21, line 8) Figure 2. Phylogenetic relationships of RBOHs. The phylogenetic tree (circle a) is constructed, after 7 members with one or two RBOH-characteristic motifs were excluded from the 134 identified RBOHs. Six subgroups are indicated with I–VI, respectively. Triangles in red, blue, purple, and green color indicate domains of NADPH_Ox, Ferric_reduct, FAD_binding_8, and NAD_binding_6, respectively. Circles b, c, d, and e are composed of four above mentioned domains within each RBOHs, respectively. Circles at the individual nodes represent bootstrap support.]

3. [The author mentioned in the Method that BLASTP was used as the scanning tool (P32, line 39), then the author must have blasted against the protein sequences from annotated genomes. If so, the impact of incomplete annotations should be considered. In P29, line 56-57, “When the identification of RBOH members was carried out, all genome sequences have been scanned whether they have annotation or not”, indicating that the author had considered this problem, but how this was done should be described in the Methods as well.]

[Thank you for your valuable suggestion. As you mentioned in the comments (In P29, line 56-57, “When the identification of RBOH members was carried out, all genome sequences have been scanned whether they have annotation or not”), we just mean that BLASTP was performed by blasting against full set of genome sequences. Due to blasting against the protein sequences from annotated genomes, both the complete and incomplete annotations have been considered. And the sequences without any annotations were filtered out from the retrieval. We have revised this description in the Material and methods section.

(Material and methods section, page 11, line 21) Due to blasting

against the protein sequences from annotated genomes, both the complete and incomplete annotations have been considered. And the sequences without any annotations were filtered out from the retrieval.]

Obediently yours,

Dahui Li

College of Life Science,
Anhui Agricultural University,
Hefei 230036, China
Telephone: +86-551-6578-6237
E-mail: dahui2@126.com